# On correlation functions for the open XXZ chain with non-longitudinal boundary fields: The case with a constraint

Giuliano Niccoli[1*] and Véronique Terras[2†]

**1** Univ. Lyon, Ens de Lyon, Univ Claude Bernard, CNRS,
Laboratoire de Physique, F-69342 Lyon, France
**2** Université Paris-Saclay, CNRS, LPTMS, 91405, Orsay, France

* giuliano.niccoli@ens-lyon.fr , † veronique.terras@universite-paris-saclay.fr

## Abstract

This paper is a continuation of [1], in which a set of matrix elements of local operators was computed for the XXZ spin-1/2 open chain with a particular case of unparallel boundary fields. Here, we extend these results to the more general case in which both fields are non-longitudinal and related by one constraint, allowing for a partial description of the spectrum by usual Bethe equations. More precisely, the complete spectrum and eigenstates can be characterized within the Separation of Variables (SoV) framework. One uses here the fact that, under the constraint, a part of this SoV spectrum can be described via solutions of a usual, homogeneous, $TQ$-equation, with corresponding transfer matrix eigenstates coinciding with generalized Bethe states. We explain how to generically compute the action of a basis of local operators on such kind of states, and this under the most general boundary condition on the last site of the chain. As a result, we can compute the matrix elements of some of these basis elements in any eigenstate described by the homogenous $TQ$-equation. Assuming, following a conjecture of Nepomechie and Ravanini, that the ground state itself can be described in this framework, we obtain multiple integral representations for these matrix elements in the half-infinite chain limit, generalizing those previously obtained in the case of longitudinal boundary fields and in the case of the special boundary conditions considered in [1].

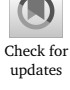

# 1 Introduction

This paper is a continuation of our previous work [1], in which a set of matrix elements of local operators on the first $m$ sites of the chain was computed for the XXZ spin-1/2 open chain with a particular case of unparallel boundary fields: there was considered the case of a fixed longitudinal boundary field on the last site of the chain, with a completely generic, a priori non-longitudinal, boundary field on the first site of the chain. Here, we instead consider the case in which both boundary fields are non-longitudinal but are related by one constraint [2] which allows for the description of a part of the spectrum and eigenstates by usual $TQ$ and Bethe equations.

There exists a large literature on open quantum spin chain [2–42]. These models have indeed numerous applications: they are useful to model out-of-equilibrium and transport properties in quantum condensed matter physics, see for instance [26]; they are also related to classical stochastic models, such as asymmetric simple exclusion models [15]. The interest in open integrable quantum spin chains lays notably on the freedom that we have in choosing its boundary conditions while keeping the system integrable.

The first results on the exact description of the Hamiltonian spectrum and eigenstates were obtained for longitudinal boundary fields, i.e. when both boundary fields are along the z-direction, in [3] within the coordinated Bethe ansatz framework, and in [4] within the algebraic Bethe ansatz (ABA) framework. In [4] a complete algebraic formalism for the study of these open spin chains was also introduced, issued from the Quantum Inverse Scattering Method (QISM) [43–45], and based on the representation theory of the reflection algebra [46]. Although the case of completely arbitrary boundary magnetic fields is a priori integrable, in the sense that one can construct, within the algebraic framework introduced in [4], a one-parameter family of commuting transfer matrices, the explicit construction of its common spectrum and eigenstates for non-longitudinal boundary fields has remained widely open for a long period. In that case, the ferromagnetic state can no longer be used as a reference state and algebraic Bethe ansatz is therefore no longer directly applicable. It was however noticed in [2] that, under a unique constraint on the six boundary parameters parametrizing the six components of the two boundary fields, one can still write a system of usual Bethe equations, which unfortunately provides in general only an incomplete description of the transfer matrix spectrum. Conjectures, based on numerical studies, were made in [13] on the fact that these Bethe equations yield the ground state at half-filling, and in [14] on the fact that one could nevertheless recover the full spectrum by combining two different sectors with different constraints related by some symmetry of the spin chain. A construction of the corresponding Bethe states via a generalization of algebraic Bethe ansatz based on the use of (a trigonometric version of) Baxter's Vertex-IRF transformation [47] was proposed in [12], see also [10,18,19].

However, it was not possible so far to compute correlation functions within ABA for this case with the constraint due to the difficulty of constructing Bethe states in both the direct and dual spaces within a common framework. As for the completely general case without the constraint, some progresses have been made more recently by the means of alternative approaches: off-diagonal Bethe ansatz [28], modified Bethe ansatz [32–37, 42], or Separation of Variables (SoV) [22, 27, 30, 31, 40]. However, the description of the spectrum proposed in these contexts does not involve usual $TQ$ and Bethe equations, but modified ones, with some additional inhomogeneous term, a description which is a priori difficult to deal with when considering the computation of physical quantities such as correlation functions and the thermodynamic limit of the model.

Hence, in the present work, we restrict our study to the case in which the boundary fields are both non-longitudinal but related by the aforementioned constraint, for which at least part of the spectrum can be described by usual Bethe equations. We suppose moreover that the boundary conditions are such that the ground state of the model is among the states which can be described in this framework, in a sector close to half-filling (see the conjectures of [13, 14]), so that it can be described in the thermodynamic limit by the same density function as in the longitudinal case with possibly some additional boundary roots, see [8, 9, 41]. We develop our study in the framework of the SoV approach [22, 25, 27, 29–31, 38–40, 48–76], pioneered by Sklyanin [48–53]. The advantages of using this approach is that it provides, on the one hand, a complete characterization of eigenvalues and eigenstates of the open spin 1/2 XXZ chain under the most general integrable boundary conditions[1] and, on the other hand, it naturally produces determinant formulae [40] for the scalar products of *separate states* [27, 29, 30, 38–40, 66, 68–71, 73, 75],[2] of which the transfer matrix eigenstates are special instances. Moreover, in this approach, the description of the spectrum in terms of Bethe equations and ABA-type construction of the eigenstates emerge explicitly as particular simplifications due to the constraint within the global approach, so that we can also make use of these results when it is more convenient to do so.

Our purpose is here to compute boundary correlation functions at zero temperature, or more precisely the mean values in the ground state of local operators on the first $m$ sites of the chain. Such kinds of results were first obtained in the periodic case [90–94], and were at the origin of a long series of impressive new and exact results concerning correlation functions [95–129]. These results could be generalized to the case of an open chain with longitudinal boundary fields in [6, 7, 20, 21]. For more general boundary fields or other types of boundary conditions, the problem has for long remained unsolved due to the aforementioned difficulties in having a good description of the spectrum and eigenstates. These limitations were overcome only recently within the SoV framework in [130] for the open XXX chain with unparallel boundary fields and in [1] for the open XXZ chain with one arbitrary non-longitudinal boundary field, the other being fixed along the z-direction (see also [131] for the XXX chain with anti-periodic boundary conditions). Here, we generalize the results of [1] to the case where both fields are non-longitudinal within the constraint, that is, we enlarge our results from the case of three free boundary parameters [1] to the case of five free boundary parameters. As in [1], we however have, due to technical difficulties related to the use of the vertex-IRF transformation, to restrict ourselves to the consideration of a subclass of matrix elements.

The content of the paper is the following. In section 2, we briefly recall the solution of the open XXZ spin 1/2 chain with non-longitudinal boundary fields: we recall the general al-

---

[1]This is in particular the case in the framework of the recent rederivation of SoV [77, 78] relying on the pure integrable structure of the models and allowing for a wide extension of its applicability even to higher rank cases [77–84]; see also [53, 56, 85, 86] for different precedent higher rank developments.

[2]This is at least the case for the rank one models. In [87], it has been shown how these types of formulae extend to the gl(3) higher rank case, see also the interesting and recent papers [88, 89].

gebraic framework, the SoV characterization of the transfer matrix spectrum and eigenstates, and the simplifications that occur when one imposes the constraint of [2] on the boundary parameters. In this case, part of the spectrum can be characterized in terms of a homogeneous $TQ$-equation, and the corresponding eigenstates – and more generally all separate states associated with a polynomial function $Q$ – can be reformulated in terms of well-identified generalized Bethe states. The latter are constructed, as in [12], from a gauged transformed version of the reflection algebra, by means of the Vertex-IRF transformation for particular values of the corresponding gauged parameters. In section 3, we explain how to decompose these generalized boundary Bethe states into generalized bulk Bethe states for any arbitrary boundary field on the last site $N$ of the chain, i.e. for any form of the associated boundary matrix which is thus a priori non-diagonal. This represents a strong generalization of the boundary-bulk decomposition derived in [1], since there was only considered the case of a particular longitudinal boundary field at site $N$, corresponding to a particular diagonal boundary matrix. Note that this boundary-bulk decomposition is a central technical point for the computation of correlation functions, since we do not know at the moment how to act directly with local operators on boundary states, but only on bulk states. Hence, the form of the boundary-bulk decomposition has to be simple enough so that we can reconstruct the result of the action in terms of boundary states. The idea is here to adjust some gauge parameters of the Vertex-IRF transformation so as to make this decomposition as simple as in the diagonal case. Quite remarkably, the choice that we have to make on the gauge parameters is compatible with the constraint, so that we in fact obtain a decomposition of separate states into bulk Bethe states on which we can act with local operators completely similarly as in [1]. The result of such action is then given in section 4. There, we consider the same basis of local operators on the first $m$ sites of the chain as in our previous work [1]. The result of the action of the elements of this basis on the boundary Bethe states/separate states obtained in section 4 is very similar, in its form, as the one derived in [1], with however different hypothesis: a more general boundary field and more constraint gauge parameters. This enables us to compute, by using the scalar product formulas derived in [40], the matrix elements of these local operators in any transfer matrix eigenstate that can be obtained by a solution of the homogenous $TQ$-equation under the constraint [2]. However, as in [1], we have to restrict ourselves to the subset of local operators that preserve the number of gauged boundary $B$-operators. The result of this action for the finite chain is presented in section 5. Finally, in section 6, we derive multiple integral representations for these matrix elements in the ground state and in the half-infinite chain limit. We there make the hypothesis (based on the conjecture of [13,14]) that the ground state can be obtained from a solution of the homogeneous $TQ$-equation close to half-filling, and can therefore be described by the same density function as in the diagonal case with possibly some additional boundary roots related to the four boundary parameters that appear in the Bethe equations.

## 2 The open spin-1/2 XXZ quantum chain

In this paper, we consider the open XXZ spin-1/2 quantum chain coupled with local external fields on sites 1 and $N$. The Hamiltonian of this model is given on $\mathcal{H} = \otimes_{n=1}^{N} \mathcal{H}_n$, $\mathcal{H}_n \simeq \mathbb{C}^2$, as

$$H = \sum_{n=1}^{N-1} \left[ \sigma_n^x \sigma_{n+1}^x + \sigma_n^y \sigma_{n+1}^y + \Delta \sigma_n^z \sigma_{n+1}^z \right] + \sum_{a \in \{x,y,z\}} \left[ h_-^a \sigma_1^a + h_+^a \sigma_N^a \right]. \qquad (2.1)$$

Here $\sigma_n^\alpha$, $\alpha \in \{x, y, z\}$ stand for the usual Pauli matrices acting on $\mathcal{H}_n$. The anisotropy $\Delta$ of the coupling is parameterized as

$$\Delta = \cosh \eta, \qquad (2.2)$$

and the boundary fields $h_\pm$ as

$$h_\pm^x = 2\kappa_\pm \sinh\eta \, \frac{\cosh\tau_\pm}{\sinh\varsigma_\pm} = \sinh\eta \, \frac{\cosh\tau_\pm}{\sinh\varphi_\pm \cosh\psi_\pm}, \tag{2.3}$$

$$h_\pm^y = 2i\kappa_\pm \sinh\eta \, \frac{\sinh\tau_\pm}{\sinh\varsigma_\pm} = i\sinh\eta \, \frac{\sinh\tau_\pm}{\sinh\varphi_\pm \cosh\psi_\pm}, \tag{2.4}$$

$$h_\pm^z = \sinh\eta \coth\varsigma_\pm = \sinh\eta \coth\varphi_\pm \tanh\psi_\pm, \tag{2.5}$$

where the two sets of boundary parameters are related by

$$\sinh\varphi_\pm \cosh\psi_\pm = \frac{\sinh\varsigma_\pm}{2\kappa_\pm}, \qquad \cosh\varphi_\pm \sinh\psi_\pm = \frac{\cosh\varsigma_\pm}{2\kappa_\pm}. \tag{2.6}$$

The spectral problem of an inhomogeneous version of the Hamiltonian (2.1) with non-longitudinal boundary fields can be solved by the quantum version of the Separation of Variables (SoV) approach [22, 27, 30, 40, 80], in the algebraic QISM framework of the representation theory of the reflection algebra [4, 46]. We here use the same definitions and notations as in our previous paper [1], in which this approach was reviewed, so that we give here only the necessary notations for the purpose of the present paper and refer to [1] for details.

## 2.1 General framework

Let us define the boundary monodromy matrix $\mathcal{U}_-(\lambda) \equiv \mathcal{U}_{-,0}(\lambda) \in \text{End}(V_0 \otimes \mathcal{H})$, $V_0 \simeq \mathbb{C}^2$, as

$$\mathcal{U}_{-,0}(\lambda) = T_0(\lambda) K_{-,0}(\lambda) \hat{T}_0(\lambda) = \begin{pmatrix} \mathcal{A}_-(\lambda) & \mathcal{B}_-(\lambda) \\ \mathcal{C}_-(\lambda) & \mathcal{D}_-(\lambda) \end{pmatrix}, \tag{2.7}$$

in terms of

$$T_0(\lambda) = R_{01}(\lambda - \xi_1 - \eta/2)\dots R_{0N}(\lambda - \xi_N - \eta/2), \tag{2.8}$$

$$\hat{T}_0(\lambda) = (-1)^N \sigma_0^y T_0^{t_0}(-\lambda) \sigma_0^y = R_{0N}(\lambda + \xi_N - \eta/2)\dots R_{01}(\lambda + \xi_1 - \eta/2), \tag{2.9}$$

and of

$$K_-(\lambda) \equiv K_{-,0}(\lambda) = K(\lambda; \varsigma_+, \kappa_+, \tau_+), \tag{2.10}$$

The bulk monodromy matrix with inhomogeneity parameters $\xi_1, \dots, \xi_N$ (2.8) is defined in terms of the 6-vertex trigonometric $R$-matrix,

$$R(\lambda) \equiv R_{12}(\lambda) = \begin{pmatrix} \sinh(\lambda+\eta) & 0 & 0 & 0 \\ 0 & \sinh\lambda & \sinh\eta & 0 \\ 0 & \sinh\eta & \sinh\lambda & 0 \\ 0 & 0 & 0 & \sinh(\lambda+\eta) \end{pmatrix} \in \text{End}(\mathbb{C}^2 \otimes \mathbb{C}^2), \tag{2.11}$$

whereas the scalar boundary matrix (2.10), which parameterizes the boundary field $h_+$, is defined as

$$K(\lambda; \varsigma, \kappa, \tau) = \frac{1}{\sinh\varsigma} \begin{pmatrix} \sinh(\lambda - \eta/2 + \varsigma) & \kappa e^\tau \sinh(2\lambda - \eta) \\ \kappa e^{-\tau} \sinh(2\lambda - \eta) & \sinh(\varsigma - \lambda + \eta/2) \end{pmatrix}. \tag{2.12}$$

Introducing also the scalar boundary matrix $K_+(\lambda) = K(\lambda + \eta; \varsigma_-, \kappa_-, \tau_-)$ parameterizing the boundary field $h_-$, we define the transfer matrice as

$$\mathcal{T}(\lambda) = \text{tr}_0\left[K_{+,0}(\lambda) T_0(\lambda) K_{-,0}(\lambda) \hat{T}_0(\lambda)\right] = \text{tr}_0\left[K_{+,0}(\lambda) \mathcal{U}_{-,0}(\lambda)\right]. \tag{2.13}$$

The latter form a one-parameter family of commuting operators on $\mathcal{H}$, and the Hamiltonian (2.1) can be obtained in terms of this transfer matrix in the homogeneous limit $\xi_1 = \xi_2 = \cdots = \xi_N = 0$ as

$$H = \frac{2(\sinh\eta)^{1-2N}}{\mathrm{tr}[K_+(\eta/2)]\,\mathrm{tr}[K_-(\eta/2)]} \frac{d}{d\lambda}\mathcal{T}(\lambda)\Big|_{\lambda=\eta/2} + \text{constant}. \tag{2.14}$$

Note that we have here used the same convention as in [1], in which we have defined $K_-$ in terms of $\varsigma_+, \kappa_+, \tau_+$ parameterizing $h_+$ and $K_+$ in terms of $\varsigma_-, \kappa_-, \tau_-$ parameterizing $h_-$.

In this framework, the transfer matrix eigenstates can be constructed by algebraic Bethe ansatz (ABA) when both boundary matrices $K_\pm$ are diagonal [4], or by Separation of Variables (SoV) otherwise [22, 27, 30, 40, 80]. In this paper we consider the situation in which the boundary matrices $K_\pm$ are both non-diagonal and in which the boundary parameters satisfy some constraint,

$$\cosh(\tau_+ - \tau_-) = \epsilon_{\varphi_+}\epsilon_{\varphi_-}\cosh(\epsilon_{\varphi_+}\varphi_+ + \epsilon_{\varphi_-}\varphi_- + \epsilon_{\psi_+}\psi_+ - \epsilon_{\psi_-}\psi_- + (N-1-2M)\eta), \tag{2.15}$$

for some $M \in \{1,\dots,N\}$ and some $\boldsymbol{\varepsilon} \equiv (\epsilon_{\varphi_+}, \epsilon_{\varphi_-}, \epsilon_{\psi_+}, \epsilon_{\psi_-}) \in \{-1,1\}^4$ such that $\epsilon_{\varphi_+}\epsilon_{\varphi_-}\epsilon_{\psi_+}\epsilon_{\psi_-} = 1$. In this case, the spectrum can be partially described by usual Bethe equations [2, 12, 31], with corresponding eigenstates constructed by SoV in terms of some polynomial function $Q(\lambda)$ of the form

$$Q(\lambda) = \prod_{j=1}^{M} \frac{\cosh(2\lambda) - \cosh(2\lambda_j)}{2} = \prod_{j=1}^{M} \left(\sinh^2\lambda - \sinh^2\lambda_j\right), \tag{2.16}$$

where $\lambda_1, \dots, \lambda_M$ are the corresponding Bethe roots. Except in the case $M = N$, such a description is not complete. However, it was conjectured in [13, 14], using numerical studies, that the solutions of the Bethe equations for some $M$ subject to the constraint (2.15) with some choice of $\boldsymbol{\varepsilon}$, together with the solutions for $M' = N-1-M$ subject to the constraint with $\boldsymbol{\varepsilon}' = -\boldsymbol{\varepsilon}$, produce the complete spectrum. We shall not discuss the validity of this conjecture in the present paper, but we will nevertheless suppose that we are in a situation in which at least the ground state can be described in this framework, which will be enough for our purpose.

## 2.2 Transfer matrix spectrum and eigenstates by SoV

Let us recall that, in the situation we consider here of non-diagonal boundary matrices $K_\pm$, and provided that the inhomogeneity parameters are generic, i.e.

$$\xi_j, \xi_j \pm \xi_k \notin \{0, -\eta, \eta\} \bmod(i\pi), \quad \forall j, k \in \{1,\dots,N\}, j \neq k, \tag{2.17}$$

the transfer matrix $\mathcal{T}(\lambda)$ has simple spectrum, and that the complete set of its left and right eigenstates can be expressed in the form of *separate states* as

$$|Q\rangle = \sum_{\mathbf{h}\in\{0,1\}^N} \prod_{n=1}^{N} \frac{Q(\xi_n^{(h_n)})}{Q(\xi_n^{(0)})}\, e^{-\sum_j h_j\xi_j}\, \widehat{V}(\xi_1^{(h_1)},\dots,\xi_N^{(h_N)})\,|\mathbf{h}\rangle, \tag{2.18}$$

$$\langle Q| = \sum_{\mathbf{h}\in\{0,1\}^N} \prod_{n=1}^{N} \left[\left(\left(\frac{\sinh(2\xi_n - 2\eta)}{\sinh(2\xi_n + 2\eta)}\frac{\mathbf{A}_{\boldsymbol{\varepsilon}}(\xi_n + \frac{\eta}{2})}{\mathbf{A}_{\boldsymbol{\varepsilon}}(-\xi_n + \frac{\eta}{2})}\right)^{h_n} \frac{Q(\xi_n^{(h_n)})}{Q(\xi_n^{(0)})}\right]$$
$$\times e^{-\sum_j h_j\xi_j}\, \widehat{V}(\xi_1^{(h_1)},\dots,\xi_N^{(h_N)})\,\langle\mathbf{h}|. \tag{2.19}$$

More precisely, the unique (up to normalization) $\mathcal{T}(\lambda)$-eigenstates associated with any given eigenvalue $\tau(\lambda)$ are given by (2.18) and (2.19) in which $Q$ is such that

$$\frac{Q(\xi_n^{(1)})}{Q(\xi_n^{(0)})} = \frac{\tau(\xi_n^{(0)})}{\mathbf{A}_{\boldsymbol{\varepsilon}}(\xi_n^{(0)})} = \frac{\mathbf{A}_{\boldsymbol{\varepsilon}}(-\xi_n^{(1)})}{\tau(\xi_n^{(1)})}, \qquad \forall n \in \{1,\dots,N\}. \tag{2.20}$$

In the above formulas,

$$\big\{\,|\mathbf{h}\rangle, \mathbf{h} \equiv (h_1, \ldots, h_N) \in \{0,1\}^N \big\} \qquad \text{and} \qquad \big\{\langle\mathbf{h}|, \mathbf{h} \equiv (h_1, \ldots, h_N) \in \{0,1\}^N \big\} \tag{2.21}$$

are SoV bases of $\mathcal{H}$ and $\mathcal{H}^*$ respectively, which can be constructed either by a generalization of Sklyanin's SoV approach [27, 30] or by the new SoV approach proposed in [80]. The normalization of such bases can be chosen so that

$$\langle \mathbf{h} \,|\, \mathbf{h}'\rangle = \delta_{\mathbf{h},\mathbf{h}'} \frac{N(\{\xi\})\, e^{2\sum_{j=1}^N h_j \xi_j}}{\widehat{V}(\xi_1^{(h_1)}, \ldots, \xi_N^{(h_N)})}, \tag{2.22}$$

where

$$N(\{\xi\}) = \widehat{V}(\xi_1, \ldots, \xi_N) \frac{\widehat{V}(\xi_1^{(0)}, \ldots, \xi_N^{(0)})}{\widehat{V}(\xi_1^{(1)}, \ldots, \xi_N^{(1)})}. \tag{2.23}$$

As in [1], we have used the notations

$$\xi_n^{(h)} = \xi_n + \eta/2 - h\eta, \qquad 1 \le n \le N, \quad h \in \{0,1\}, \tag{2.24}$$

$$\widehat{V}(x_1, \ldots, x_N) = \det_{1 \le i,j \le N}\left[\sinh^{2(j-1)} x_i\right] = \prod_{j<k}(\sinh^2 x_k - \sinh^2 x_j), \tag{2.25}$$

Moreover we have defined,

$$\mathbf{A}_{\boldsymbol{\varepsilon}}(\lambda) = (-1)^N \frac{\sinh(2\lambda + \eta)}{\sinh(2\lambda)}\, \mathbf{a}_{\boldsymbol{\varepsilon}}(\lambda)\, a(\lambda)\, d(-\lambda), \tag{2.26}$$

where

$$a(\lambda) = \prod_{n=1}^N \sinh(\lambda - \xi_n + \eta/2), \qquad d(\lambda) = \prod_{n=1}^N \sinh(\lambda - \xi_n - \eta/2), \tag{2.27}$$

and

$$\begin{aligned}
\mathbf{a}_{\boldsymbol{\varepsilon}}(\lambda) &= \frac{\sinh(\lambda - \tfrac{\eta}{2} + \epsilon_{\varphi_+}\varphi_+)\cosh(\lambda - \tfrac{\eta}{2} + \epsilon_{\psi_+}\psi_+)}{\sinh(\epsilon_{\varphi_+}\varphi_+)\cosh(\epsilon_{\psi_+}\psi_+)} \\
&\quad \times \frac{\sinh(\lambda - \tfrac{\eta}{2} + \epsilon_{\varphi_-}\varphi_-)\cosh(\lambda - \tfrac{\eta}{2} - \epsilon_{\psi_-}\psi_-)}{\sinh(\epsilon_{\varphi_-}\varphi_-)\cosh(\epsilon_{\psi_-}\psi_-)},
\end{aligned} \tag{2.28}$$

for any choice of $\boldsymbol{\varepsilon} \equiv (\epsilon_{\varphi_+}, \epsilon_{\varphi_-}, \epsilon_{\psi_+}, \epsilon_{\psi_-}) \in \{-1,1\}^4$ such that $\epsilon_{\varphi_+}\epsilon_{\varphi_-}\epsilon_{\psi_+}\epsilon_{\psi_-} = 1$.

Note that the new SoV construction proposed in [80] (see Section 3.1.2 of our previous work [1] for a summary of this construction in our present notations) is more general: it is valid as long as the two boundary matrices $K_+$ and $K_-$ are not both proportional to the identity. The generalization of Sklyanin's SoV approach proposed in [30] (see Section 3.1.1 of [1]) is instead restricted to the case in which at least one boundary matrix (the boundary matrix $K_-$ with our conventions) is non-diagonal. It involves a generalized gauge transformation of the reflection algebra,

$$\begin{aligned}
\mathcal{U}_-(\lambda|\alpha,\beta) &= S_0^{-1}(\eta/2 - \lambda|\alpha,\beta)\, \mathcal{U}_-(\lambda)\, S_0(\lambda - \eta/2|\alpha,\beta) \\
&= \begin{pmatrix} \mathcal{A}_-(\lambda|\alpha,\beta) & \mathcal{B}_-(\lambda|\alpha,\beta) \\ \mathcal{C}_-(\lambda|\alpha,\beta) & \mathcal{D}_-(\lambda|\alpha,\beta) \end{pmatrix},
\end{aligned} \tag{2.29}$$

given by a trigonometric version of Baxter's Vertex-IRF transformation,

$$S(\lambda|\alpha,\beta) = \begin{pmatrix} e^{\lambda - \eta(\beta+\alpha)} & e^{\lambda + \eta(\beta-\alpha)} \\ 1 & 1 \end{pmatrix}, \tag{2.30}$$

with a particular choice[3] of the two parameters $\alpha$ and $\beta$ as

$$\eta\alpha = -\tau_- + \frac{\epsilon'_- - \epsilon_-}{2}(\varphi_- - \psi_-) - \frac{\epsilon_- + \epsilon'_-}{4}i\pi + ik\pi \quad \mod 2i\pi, \tag{2.31}$$

$$\eta\beta = \frac{\epsilon_- + \epsilon'_-}{2}(\varphi_- - \psi_-) + \frac{2 + \epsilon_- - \epsilon'_-}{4}i\pi + ik\pi \quad \mod 2i\pi, \tag{2.32}$$

for $\epsilon_-, \epsilon'_- \in \{1, -1\}$ and $k \in \mathbb{Z}$. Since it is not necessary for our purpose and was already recalled in our previous work [1], we do not recall the details of this construction here and refer instead the reader to [1]. We also refer to [1] for general properties of the gauged transformed algebra generated by the operator entries of (2.29).

## 2.3 Reformulation of the spectrum and eigenstates

One of the difficulties of the SoV approach is that it provides a priori a characterisation of the transfer spectrum and eigenstates in terms of discrete equations of the form (2.20). Such a characterisation is not convenient for the consideration of the physical model at the homogeneous limit $\xi_1 = \xi_2 = \cdots = \xi_N = 0$ and the computation of physical quantities such as the correlation functions. Hence, this characterisation has been reformulated in [31] in terms of functional $TQ$-equations, leading notably to usual Bethe equations in the cases in which the constraint (2.15) holds.

Let us first introduce the following notations:

$$\mathfrak{f}^{(r)}_{\boldsymbol{\varepsilon}} \equiv \mathfrak{f}^{(r)}_{\boldsymbol{\varepsilon}}(\tau_+, \tau_-, \varphi_+, \varphi_-, \psi_+, \psi_-) = \frac{2\kappa_+ \kappa_-}{\sinh\varsigma_+ \sinh\varsigma_-}\Big[\cosh(\tau_+ - \tau_-)$$
$$- \epsilon_{\varphi_+}\epsilon_{\varphi_-}\cosh(\epsilon_{\varphi_+}\varphi_+ + \epsilon_{\varphi_-}\varphi_- + \epsilon_{\psi_+}\psi_+ - \epsilon_{\psi_-}\psi_- + (N - 1 - 2r)\eta)\Big]. \tag{2.33}$$

Moreover, we denote with $\Sigma_Q^M$ the set of $Q(\lambda)$ polynomials in $\cosh(2\lambda)$ of degree $M$ of the form (2.16) with

$$\cosh(2\lambda_j) \neq \cosh(2\xi_n^{(h)}), \quad \forall (j, n, h) \in \{1, \ldots, M\} \times \{1, \ldots, N\} \times \{0, 1\}. \tag{2.34}$$

**Proposition 2.1** ([31,40], see also [1]). *Let the two boundary matrices be not both proportional to the identity matrix and the inhomogeneity parameters be generic (2.17).*

1. *Let us suppose that, for a given $\boldsymbol{\varepsilon} \equiv (\epsilon_{\varphi_+}, \epsilon_{\varphi_-}, \epsilon_{\psi_+}, \epsilon_{\psi_-}) \in \{-1, 1\}^4$ such that $\epsilon_{\varphi_+}\epsilon_{\varphi_-}\epsilon_{\psi_+}\epsilon_{\psi_-} = 1$,*

$$\forall r \in \{0, \ldots, N-1\}, \qquad \mathfrak{f}^{(r)}_{\boldsymbol{\varepsilon}}(\tau_+, \tau_-, \varphi_+, \varphi_-, \psi_+, \psi_-) \neq 0. \tag{2.35}$$

*Then, the transfer matrix $\mathcal{T}(\lambda)$ is diagonalizable with simple spectrum, and the set $\Sigma_{\mathcal{T}}$ of its eigenvalues is given by the set of entire functions $\tau(\lambda)$ such that there exists a polynomial $Q(\lambda) \in \Sigma_Q^N$ satisfying with $\tau(\lambda)$ the following TQ-equation with inhomogeneous term:*

$$\tau(\lambda)Q(\lambda) = \mathbf{A}_{\boldsymbol{\varepsilon}}(\lambda)Q(\lambda - \eta) + \mathbf{A}_{\boldsymbol{\varepsilon}}(-\lambda)Q(\lambda + \eta)$$
$$+ \mathfrak{f}^{(N)}_{\boldsymbol{\varepsilon}} a(\lambda)a(-\lambda)d(\lambda)d(-\lambda)\big[\cosh^2(2\lambda) - \cosh^2\eta\big]. \tag{2.36}$$

*Moreover, in that case, the corresponding $Q(\lambda) \in \Sigma_Q^N$ satisfying (2.36) with $\tau(\lambda)$ is unique, and the unique (up to an overall normalization factor) left and right $\mathcal{T}(\lambda)$ eigenstates can be expressed as (2.18)-(2.19).*

---

[3]This choice ensures that the transfer matrix (2.13) can be written only in terms of the elements $\mathcal{A}_-(\lambda|\alpha, \beta - 1)$ and $\mathcal{D}_-(\lambda|\alpha, \beta + 1)$ of the gauged transformed monodromy matrix (2.29), see Proposition 2.2 of [1].

2. *Let us suppose that*

$$\frac{\kappa_+ \kappa_-}{\sinh \varsigma_+ \sinh \varsigma_-} = 0 \, . \tag{2.37}$$

*Then, the transfer matrix $\mathcal{T}(\lambda)$ is diagonalizable with simple spectrum, and the set $\Sigma_{\mathcal{T}}$ of its eigenvalues is given by the set of entire functions $\tau(\lambda)$ such that there exists a polynomial $Q(\lambda) \in \cup_{n=0}^{N} \Sigma_Q^n$ satisfying with $\tau(\lambda)$ the following homogeneous TQ-equation:*

$$\tau(\lambda) Q(\lambda) = \mathbf{A}_{\boldsymbol{\varepsilon}}(\lambda) Q(\lambda - \eta) + \mathbf{A}_{\boldsymbol{\varepsilon}}(-\lambda) Q(\lambda + \eta) \, , \tag{2.38}$$

*for a given $\boldsymbol{\varepsilon} \equiv (\epsilon_{\varphi_+}, \epsilon_{\varphi_-}, \epsilon_{\psi_+}, \epsilon_{\psi_-}) \in \{-1, 1\}^4$ and $\epsilon_{\varphi_+} \epsilon_{\varphi_-} \epsilon_{\psi_+} \epsilon_{\psi_-} = 1$, Moreover, in that case, the corresponding $Q(\lambda) \in \cup_{n=0}^{N} \Sigma_Q^n$ satisfying (2.38) with $\tau(\lambda)$ is unique, and the unique (up to an overall normalization factor) left and right $\mathcal{T}(\lambda)$ eigenstates can be expressed as (2.18)-(2.19).*

3. *Let us suppose that the condition (2.15) is satisfied for a given $M \in \{0, \ldots, N-1\}$ and a given choice of $\boldsymbol{\varepsilon} \equiv (\epsilon_{\varphi_+}, \epsilon_{\varphi_-}, \epsilon_{\psi_+}, \epsilon_{\psi_-}) \in \{-1, 1\}^4$ such that $\epsilon_{\varphi_+} \epsilon_{\varphi_-} \epsilon_{\psi_+} \epsilon_{\psi_-} = 1$. Then, the transfer matrix is diagonalizable with simple spectrum, and any entire function $\tau(\lambda)$ such that there exists $Q(\lambda) \in \Sigma_Q^M$ satisfying the homogeneous TQ-equation (2.38) is an eigenvalue of the transfer matrix $\mathcal{T}(\lambda)$ (we write $\tau(\lambda) \in \Sigma_{\mathcal{T}}$). Moreover, in that case, the corresponding $Q(\lambda) \in \Sigma_Q^M$ satisfying (2.38) with $\tau(\lambda)$ is unique, and the unique (up to an overall normalization factor) left and right $\mathcal{T}(\lambda)$ eigenstates can be expressed as (2.18)-(2.19).*

*Remark* 1. In the formulation of this proposition, the bases (2.21) used in (2.18)-(2.19) can be either the SoV bases constructed via the new SoV approach proposed in [80], or the bases constructed via the generalized Sklyanin's approach, provided that the latter approach is applicable (we recall that its range of validity is more restricted).

In our previous work [1], we considered the case 2, which corresponds to the situation in which (at least) one of the two boundary matrices is diagonal. Here instead we want to consider the case 3, in which both boundary matrices are non-diagonal and satisfy the constraint (2.15). Note that the characterization of the spectrum and eigenstates provided by this proposition in the case 3 is not complete, contrary to what happens in cases 1 and 2. As explained in [31, 40], a part of the spectrum is then given by solutions of the inhomogeneous *TQ*-equation (2.36). It can also be noticed that, if the constraint (2.15) is satisfied for a given $M$ and a given $\boldsymbol{\varepsilon}$, then it is also satisfied for $M' = N - 1 - M$ and $\boldsymbol{\varepsilon}' = -\boldsymbol{\varepsilon}$. As mentioned above, it was conjectured in [13, 14] that the solutions for $M, \boldsymbol{\varepsilon}$ together with the solutions for $M', \boldsymbol{\varepsilon}'$ produce the complete spectrum.

Under such a reformulation of the spectrum, any separate state constructed as in (2.18)-(2.19) within the generalization of Sklyanin's SoV approach can be expressed as a generalized Bethe state in the framework of the generalized gauge transformed reflection algebra given by (2.29), see (3.59)-(3.62) of [1]. Moreover, under certain conditions, the corresponding generalized reference state can be explicitly identified.

Let us introduce, as in [1], the notations

$$| \eta, x \rangle \equiv \otimes_{n=1}^{N} \begin{pmatrix} e^{-(n-N+x)\eta - \xi_n} \\ 1 \end{pmatrix}_n \, , \tag{2.39}$$

$$\widehat{\mathcal{B}}_-(\lambda | \alpha - \beta) = \sinh(\eta\beta) \, e^{-\eta\beta} \, e^{-(\lambda - \eta/2)} \, \mathcal{B}_-(\lambda | \alpha, \beta) \, , \tag{2.40}$$

and

$$\begin{aligned}
\underline{\widehat{\mathcal{B}}}_{-,M}(\{\lambda_i\}_{i=1}^{M} | \alpha - \beta + 1) &= \widehat{\mathcal{B}}_-(\lambda_1 | \alpha - \beta + 1) \cdots \widehat{\mathcal{B}}_-(\lambda_M | \alpha - \beta + 2M - 1) \\
&= \prod_{j=1 \to M} \widehat{\mathcal{B}}_-(\lambda_j | \alpha - \beta + 2j - 1) \, .
\end{aligned} \tag{2.41}$$

Then we can state the following proposition, which is just a reformulation of Proposition 3.4 of [1]:

**Proposition 2.2.** *Let us suppose that, for a given $M \in \{1, \dots, N\}$ and given $\epsilon_{\varphi_+}, \epsilon_{\varphi_-} \in \{+1, -1\}$,*

$$\tau_+ - \tau_- = -\epsilon_{\varphi_+}(\varphi_+ + \psi_+) - \epsilon_{\varphi_-}(\varphi_- - \psi_-) - (N - 1 - 2M)\eta + \frac{1 - \epsilon_{\varphi_+}\epsilon_{\varphi_-}}{2} i\pi \quad \mod 2i\pi. \quad (2.42)$$

*Let us moreover suppose that the generalized Sklyanin's approach is applicable, with $\alpha$ and $\beta$ fixed in terms of the boundary parameters $\varphi_-, \psi_-$ and of $\epsilon_{\varphi_-}$ as*

$$\eta\alpha = -\tau_- + \frac{\epsilon_{\varphi_-} - \epsilon_-}{2}(\varphi_- - \psi_-) - \frac{\epsilon_- + \epsilon_{\varphi_-}}{4} i\pi + ik\pi \quad \mod 2i\pi, \quad (2.43)$$

$$\eta\beta = \frac{\epsilon_- + \epsilon_{\varphi_-}}{2}(\varphi_- - \psi_-) + \frac{2 + \epsilon_- - \epsilon_{\varphi_-}}{4} i\pi + ik\pi \quad \mod 2i\pi, \quad (2.44)$$

*for any $\epsilon_- \in \{1, -1\}$, $k \in \mathbb{Z}$. Then, there exists a constant $c^{(R)}_{M,ref}$ such that, for any $Q \in \Sigma^M_Q$ with roots $\lambda_1, \dots, \lambda_M$ labelled as in (2.16), the separate state (2.18) constructed within the generalized Sklyanin's SoV approach with $\alpha, \beta$ given by (2.43)-(2.44) can be written as*

$$|Q\rangle = c^{(R)}_{M,ref} \, \widehat{\underline{\mathcal{B}}}_{-,M}(\{\lambda_i\}^M_{i=1} | \alpha - \beta + 1) | \eta, \alpha + \beta + N - 2M - 1 \rangle. \quad (2.45)$$

Note that the ABA rewriting of (2.45) is valid for any separate state constructed within the Sklyanin's approach, and not only for eigenstates (i.e. $Q$ in (2.45) does not need to satisfy the $TQ$-equation). Such a rewriting does however not hold in general for separate states constructed via the new SoV approach, except for eigenstates, see [1] for more details.

# 3 Decomposition of boundary states into bulk ones

To compute correlation functions, we need to be able to act with the local operators on the transfer matrix eigenstates. In the bulk case [93, 94], this could be done thanks to the solution of the quantum inverse problem [93, 132, 133], i.e. of the expression of local operators in terms of the generators of the Yang-Baxter algebra. In the absence of a solution of the quantum inverse problem directly in terms of the boundary algebra, we shall use the same strategy as in [1, 20]: decompose the boundary eigenstates in terms of bulk states on which we are able to act by using the bulk inverse problem, see [1] for more details on this procedure. To this purpose, the representation of eigenstates, and more generally of separate states (in the Sklyanin's framework) in terms of generalized Bethe states of Proposition 2.2, is particularly convenient. In this section, we explain how to decompose these generalized Bethe states into generalized (gauged transformed) bulk Bethe states for any boundary matrix $K_-$.

## 3.1 Boundary-bulk decomposition of the monodromy matrix

The idea is, as in [20] to use the decomposition of the boundary monodromy matrix into the bulk one. However, as in [1], instead of using directly (2.7) giving $\mathcal{U}_-$ in terms of $T$ (2.8), we prefer for technical reasons to use the decomposition

$$\mathcal{U}_-(\lambda) = \hat{M}(-\lambda) K_-(\lambda) M(-\lambda), \quad (3.1)$$

in terms of

$$M(\lambda) = \bar{R}_{0N}(\lambda - \xi_N + \eta/2) \dots \bar{R}_{01}(\lambda - \xi_1 + \eta/2) = \begin{pmatrix} A(\lambda) & B(\lambda) \\ C(\lambda) & D(\lambda) \end{pmatrix}, \quad (3.2)$$

$$= (-1)^N \hat{T}(-\lambda), \quad (3.3)$$

and

$$\hat{M}(\lambda) = (-1)^N \sigma_0^y M^{t_0}(-\lambda) \sigma_0^y = \bar{R}_{01}(\lambda + \xi_1 + \eta/2)\ldots\bar{R}_{0N}(\lambda + \xi_N + \eta/2) \quad (3.4)$$

$$= (-1)^N T(-\lambda), \quad (3.5)$$

in which

$$\bar{R}_{12}(\lambda) = \begin{pmatrix} \sinh(\lambda - \eta) & 0 & 0 & 0 \\ 0 & \sinh\lambda & -\sinh\eta & 0 \\ 0 & -\sinh\eta & \sinh\lambda & 0 \\ 0 & 0 & 0 & \sinh(\lambda - \eta) \end{pmatrix} = -R_{12}(-\lambda), \quad (3.6)$$

is the $R$-matrix which corresponds to a change of parameter $\eta \to \bar{\eta} = -\eta$ with respect to (2.11). Using the Vertex-IRF transformation (2.29)-(2.30), we can then write, for any choices of the gauge parameters $\alpha, \beta, \gamma, \delta, \gamma', \delta'$ such that the corresponding matrices (2.30) are invertible,

$$\mathcal{U}_-(\lambda|\alpha,\beta) = \hat{M}(-\lambda|(\gamma,\delta),(\alpha,\beta)) K_-(\lambda|(\gamma,\delta),(\gamma',\delta')) M(-\lambda|(\gamma',\delta'),(\alpha,\beta)), \quad (3.7)$$

in which we have defined

$$M(\lambda|(\alpha,\beta),(\gamma,\delta)) = S^{-1}(-\eta/2 - \lambda|\alpha,\beta) M(\lambda) S(-\eta/2 - \lambda|\gamma,\delta)$$
$$= \begin{pmatrix} A(\lambda|(\alpha,\beta),(\gamma,\delta)) & B(\lambda|(\alpha,\beta),(\gamma,\delta)) \\ C(\lambda|(\alpha,\beta),(\gamma,\delta)) & D(\lambda|(\alpha,\beta),(\gamma,\delta)) \end{pmatrix}, \quad (3.8)$$

$$\hat{M}(\lambda|(\alpha,\beta),(\gamma,\delta)) = S^{-1}(\lambda + \eta/2|\gamma,\delta) \hat{M}(\lambda) S(\lambda + \eta/2|\alpha,\beta)$$
$$= (-1)^N \frac{\det S(\lambda + \eta/2|\alpha,\beta)}{\det S(\lambda + \eta/2|\gamma,\delta)} \sigma_0^y M^{t_0}(-\lambda|(\alpha-1,\beta),(\gamma-1,\delta)) \sigma_0^y, \quad (3.9)$$

and

$$K_-(\lambda|(\gamma,\delta),(\gamma',\delta')) = S^{-1}(\eta/2 - \lambda|\gamma,\delta) K_-(\lambda) S(\lambda - \eta/2|\gamma',\delta'), \quad (3.10)$$

see [1] for more details. As in [1], we shall also use the notations

$$M(\lambda|(\alpha,\beta),(\gamma,\delta)) = \frac{e^{\eta(\alpha+1/2)}}{2\sinh\eta\beta} \begin{pmatrix} A(\lambda|\alpha-\beta,\gamma+\delta) & B(\lambda|\alpha-\beta,\gamma-\delta) \\ C(\lambda|\alpha+\beta,\gamma+\delta) & D(\lambda|\alpha+\beta,\gamma-\delta) \end{pmatrix}. \quad (3.11)$$

highlighting the fact that the entries of (3.11) depend in fact only on two combinations $\alpha \pm \beta$ and $\gamma \pm \delta$. In terms of these matrix elements, the relation (3.7) can be rewritten as

$$\mathcal{U}_-(\lambda|\alpha,\beta) = \frac{(-1)^N e^{\eta(\gamma'+\alpha)}}{4\sinh\eta\delta'\sinh\eta\beta} \begin{pmatrix} D(\lambda|\gamma+\delta-1,\alpha-\beta-1) & -B(\lambda|\gamma-\delta-1,\alpha-\beta-1) \\ -C(\lambda|\gamma+\delta-1,\alpha+\beta-1) & A(\lambda|\gamma-\delta-1,\alpha+\beta-1) \end{pmatrix}$$
$$\times K_-(\lambda|(\gamma,\delta),(\gamma',\delta')) \begin{pmatrix} A(-\lambda|\gamma'-\delta',\alpha+\beta) & B(-\lambda|\gamma'-\delta',\alpha-\beta) \\ C(-\lambda|\gamma'+\delta',\alpha+\beta) & D(-\lambda|\gamma'+\delta',\alpha-\beta) \end{pmatrix}, \quad (3.12)$$

This relation was used in [1] to express the generalized boundary creation operators $\widehat{\mathcal{B}}_-(\lambda_1|\alpha - \beta)$ into gauged bulk operators, and then the generalized boundary Bethe states into gauged bulk Bethe states. This was done there in a particular case in which the boundary matrix $K_-$, and hence its gauged counterpart (3.10), was diagonal, so that the boundary-bulk decomposition obtained in [1] was particularly simple: it was just similar to the one used in [20]. Here we explain how to generalize such formulas to the case of any general boundary matrix $K_-$. As shown below, this can be done by choosing adequately the internal gauge parameters $\gamma, \delta, \gamma', \delta'$ in (3.7).

**Proposition 3.1.** *Let us fix the internal gauge parameters as*

$$\gamma = \gamma', \qquad \eta\gamma = -\tau_+ + \epsilon_+ i\pi/2, \tag{3.13}$$

$$\delta = \delta', \qquad \eta\delta = -\epsilon_+(\varphi_+ + \psi_+) - i\pi/2, \tag{3.14}$$

*with $\epsilon_+ = \pm 1$. Then, for any choice of the external gauge parameter $\alpha - \beta$, the following boundary bulk decomposition holds:*

$$
\widehat{\mathcal{B}}_-(\lambda|\alpha - \beta) = \frac{(-1)^N e^{\eta(\gamma + \alpha - \beta)}}{4\sinh\eta(\delta + 1)} \frac{\sinh(2\lambda - \eta)}{\sinh 2\lambda}
$$
$$
\times \big[ A_{\epsilon_+}(\lambda) B(-\lambda|\gamma - \delta - 1, \alpha - \beta - 1) D(\lambda|\gamma + \delta, \alpha - \beta)
$$
$$
- A_{\epsilon_+}(-\lambda) B(\lambda|\gamma - \delta - 1, \alpha - \beta - 1) D(-\lambda|\gamma + \delta, \alpha - \beta)\big], \tag{3.15}
$$

*where*

$$
A_{\epsilon_+}(\lambda) = \frac{\sinh(\lambda - \frac{\eta}{2} + \epsilon_+\varphi_+)\cosh(\lambda - \frac{\eta}{2} + \epsilon_+\psi_+)}{\sinh(\epsilon_+\varphi_+)\cosh(\epsilon_+\psi_+)}. \tag{3.16}
$$

*Proof.* Let us recall that the gauged boundary matrix $K_-(\lambda|(\gamma, \delta), (\gamma', \delta'))$ can be made diagonal, identically with respect to $\lambda$, by fixing $\gamma, \gamma'$ and $\delta, \delta'$ such that

$$\gamma = \gamma', \qquad \delta = \delta', \tag{3.17}$$

and that

$$\frac{\kappa_+}{\sinh\zeta_+}\big[\sinh(\eta(\gamma + \delta) + \tau_+) + \sinh(\varphi_+ + \psi_+)\big] = 0, \tag{3.18}$$

$$\frac{\kappa_+}{\sinh\zeta_+}\big[\sinh(\eta(\gamma - \delta) + \tau_+) + \sinh(\varphi_+ + \psi_+)\big] = 0. \tag{3.19}$$

With the choice (3.13)-(3.14) these conditions are satisfied, and we have

$$\big[K_-(\lambda|(\gamma, \delta), (\gamma, \delta))\big]_{11} = e^{\lambda - \eta/2} A_{\epsilon_+}(\lambda), \tag{3.20}$$

$$\big[K_-(\lambda|(\gamma, \delta), (\gamma, \delta))\big]_{22} = e^{\lambda - \eta/2} A_{-\epsilon_+}(\lambda), \tag{3.21}$$

so that,

$$
\widehat{\mathcal{B}}_-(\lambda|\alpha - \beta) = (-1)^N \frac{e^{-\lambda + \eta/2 + \eta(\gamma + \alpha - \beta)}}{4\sinh\eta\delta}
$$
$$
\times \Big\{ [K_-(\lambda|(\gamma, \delta), (\gamma, \delta))]_{11} D(\lambda|\gamma + \delta - 1, \alpha - \beta - 1) B(-\lambda|\gamma - \delta, \alpha - \beta)
$$
$$
- [K_-(\lambda|(\gamma, \delta), (\gamma, \delta))]_{22} B(-\lambda|\gamma - \delta - 1, \alpha - \beta - 1) D(-\lambda|\gamma + \delta, \alpha - \beta)\Big\}. \tag{3.22}
$$

By using the commutation relation

$$
D(\lambda|\gamma + \delta - 1, \alpha - \beta - 1) B(-\lambda|\gamma - \delta, \alpha - \beta)
$$
$$
= \frac{\sinh(\eta\delta)\sinh(2\lambda - \eta)}{\sinh(2\lambda)\sinh(\eta(\delta + 1))} B(-\lambda|\gamma - \delta - 1, \alpha - \beta - 1) D(\lambda|\gamma + \delta, \alpha - \beta)
$$
$$
+ \frac{\sinh\eta \sinh(2\lambda + \delta\eta)}{\sinh(2\lambda)\sinh(\eta(\delta + 1))} B(\lambda|\gamma - \delta - 1, \alpha - \beta - 1) D(-\lambda|\gamma + \delta, \alpha - \beta), \tag{3.23}
$$

one gets the result. $\qquad\square$

## 3.2 Decomposition of gauged boundary states into bulk ones

From the previous result one gets the following one on gauged boundary states:

**Proposition 3.2.** *Let $\{\lambda_1, \ldots, \lambda_M\}$ be an arbitrary set of spectral parameters and $\alpha, \beta$ be arbitrary gauge parameters. Then, for $\gamma$ and $\delta$ fixed as in (3.13)-(3.14) for a given $\epsilon_+ = \pm 1$, we have the following boundary-bulk decompositions for the gauged boundary Bethe state given by the action of (2.41) on the gauge reference state $|\eta, \gamma + \delta\rangle$ (2.39):*

$$\underline{\widehat{\mathcal{B}}}_{-,M}(\{\lambda_i\}_{i=1}^M | \alpha - \beta + 1) | \eta, \gamma + \delta\rangle = h_M(\gamma - \delta, \alpha - \beta, \gamma + \delta) \sum_{\sigma_1 = \pm 1, \ldots, \sigma_M = \pm 1} H_{\sigma_1, \ldots, \sigma_M}(\{\lambda_i\}_{i=1}^M)$$

$$\times B(\lambda_M^{(\sigma)} | \gamma - \delta - 1, \alpha - \beta) \cdots B(\lambda_1^{(\sigma)} | \gamma - \delta - M, \alpha - \beta + M - 1) | \eta, \gamma + \delta + M\rangle, \quad (3.24)$$

*with*

$$H_{\sigma_1, \ldots, \sigma_M}(\{\lambda_i\}_{i=1}^M) \equiv H_{\sigma_1, \ldots, \sigma_M}(\{\lambda_i\}_{i=1}^M | \epsilon_+ \varphi_+, \epsilon_+ \psi_+)$$

$$= \prod_{n=1}^M \left[ \sigma_n a(-\lambda_n^{(\sigma)}) A_{\epsilon_+}(-\lambda_n^{(\sigma)}) \frac{\sinh(2\lambda_n - \eta)}{\sinh 2\lambda_n} \right] \prod_{1 \le a < b \le M} \frac{\sinh(\lambda_a^{(\sigma)} + \lambda_b^{(\sigma)} + \eta)}{\sinh(\lambda_a^{(\sigma)} + \lambda_b^{(\sigma)})}, \quad (3.25)$$

$$h_M(\gamma - \delta, \alpha - \beta, \gamma + \delta) = (-1)^{MN} e^{M\eta \frac{\gamma - \delta + \alpha - \beta + N}{2}} \prod_{j=1}^M \frac{\sinh(\eta \frac{\alpha - \beta - \gamma - \delta + N - 1 + 2j}{2})}{2 \sinh(\eta(\delta + j))}, \quad (3.26)$$

*where we have used the notation $\lambda_n^{(\sigma)} \equiv \sigma_n \lambda_n$ for $n \in \{1, \ldots, M\}$.*

*Proof.* As in the particular case considered in [1], we proceed by induction on $M$. The result clearly holds for $M = 1$ from (3.15). Let us now suppose that it holds for a given $M$. Then, we have that by Proposition 3.1 that

$$\widehat{\mathcal{B}}_-(\lambda_{M+1} | \alpha - \beta + 1) \underline{\widehat{\mathcal{B}}}_{-,M}(\{\lambda_i\}_{i=1}^M | \alpha - \beta + 3) | \eta, \gamma + \delta\rangle$$

$$= \frac{(-1)^N e^{\eta(\gamma + \alpha - \beta + 1)}}{4 \sinh \eta(\delta + 1)} \frac{\sinh(2\lambda_{M+1} - \eta)}{\sinh 2\lambda_{M+1}} h_M(\gamma - \delta, \alpha - \beta + 2, \gamma + \delta) \sum_{\sigma_1 = \pm 1, \ldots, \sigma_M = \pm 1} H_{\sigma_1, \ldots, \sigma_M}(\{\lambda_i\}_{i=1}^M)$$

$$\times \sum_{\sigma_{M+1} = \pm} -\sigma_{M+1} A_{\epsilon_+}(-\lambda_{M+1}^{(\sigma)}) B(\lambda_{M+1}^{(\sigma)} | \gamma - \delta - 1, \alpha - \beta) D(-\lambda_{M+1}^{(\sigma)} | \gamma + \delta, \alpha - \beta + 1)$$

$$\times B(\lambda_M^{(\sigma)} | \gamma - \delta - 1, \alpha - \beta + 2) \cdots B(\lambda_1^{(\sigma)} | \gamma - \delta - M, \alpha - \beta + M + 1) | \eta, \gamma + \delta + M\rangle. \quad (3.27)$$

Now it is easy to show that the direct action of $D(-\lambda_{M+1}^{(\sigma)} | \gamma + \delta, \alpha - \beta - 1)$ produces a terms which satisfies the induction:

$$D(-\lambda_{M+1}^{(\sigma)} | \gamma + \delta, \alpha - \beta - 1) B(\lambda_M^{(\sigma)} | \gamma - \delta - 1, \alpha - \beta + 2) \cdots$$

$$\cdots B(\lambda_1^{(\sigma)} | \gamma - \delta - M, \alpha - \beta + M + 1) | \eta, \gamma + \delta + M\rangle$$

$$\overset{\text{Direct Action}}{=} \frac{e^{-(\alpha - \beta + M - 1)\eta} - e^{-(\gamma + \delta + M - N)\eta}}{e^{\eta/2}} \frac{a(-\lambda_{M+1}^{(\sigma)}) \sinh \eta(\delta + 1)}{\sinh \eta(\delta + M + 1)} \prod_{a=1}^{M-1} \frac{\sinh(\lambda_M^{(\sigma)} + \lambda_a^{(\sigma)} + \eta)}{\sinh(\lambda_M^{(\sigma)} + \lambda_a^{(\sigma)})}$$

$$\times B(\lambda_M^{(\sigma)} | \gamma - \delta - 2, \alpha - \beta + 1) \cdots B(\lambda_1^{(\sigma)} | \gamma - \delta - M - 1, \alpha - \beta + M) | \eta, \gamma + \delta + M + 1\rangle, \quad (3.28)$$

which produces the left hand side of (3.24) for $M + 1$.

It therefore remains to show that the indirect action of $D(-\lambda_{M+1}^{(\sigma)} | \gamma + \delta, \alpha - \beta - 1)$ produces terms which sum to zero to complete the proof of the induction. Let us consider the indirect

action which results in the exchange of $\pm\lambda_a$, for $a \leq M$, with $\pm\lambda_{M+1}$ in the monomial of gauged bulk $B$-operators, i.e. into the following vector:

$$B(\lambda_{M+1}|\gamma-\delta-1,\alpha-\beta)\,B(-\lambda_{M+1}|\gamma-\delta-2,\alpha-\beta+1)$$
$$\times \prod_{j=1\to M-1} B(\mu_{M-j}|\gamma-\delta-2-j,\alpha-\beta+1+j)|\,\eta,\gamma+\delta+M+1\rangle,\quad (3.29)$$

in which $\{\mu_1,\ldots,\mu_{M-1}\} \equiv \{\lambda_1\ldots,\lambda_M\} \setminus \{\lambda_a\}$. Then the coefficient in front of this vector contains the following factor:

$$\sum_{\sigma_a=\pm 1,\sigma_{M=1}=\pm 1} \sigma_{M+1}\,\sigma_a\,\mathrm{A}_{\epsilon_+}(-\lambda_{M+1}^{(\sigma)})\,\mathrm{A}_{\epsilon_+}(-\lambda_a^{(\sigma)})\,\frac{\sinh(\lambda_{M+1}^{(\sigma)}+\lambda_a^{(\sigma)}-\eta(\delta+1))}{\sinh(\lambda_{M+1}^{(\sigma)}+\lambda_a^{(\sigma)})}$$
$$= -\sinh(2\lambda_{M+1})\sinh(2\lambda_a)\,\frac{\cosh(\epsilon_+(\varphi_++\psi_+)+\eta\delta)\cosh(\epsilon_+(\varphi_++\psi_+)-\eta)}{\sinh^2(\epsilon_+\varphi_+)\cosh^2(\epsilon_+\psi_+)},\quad (3.30)$$

which is zero, being

$$\cosh(\epsilon_+(\varphi_++\psi_+)+\eta\delta)=0 \qquad (3.31)$$

for $\delta$ given by (3.14). $\qquad\square$

# 4 Action of local operators on boundary separate states

Using the boundary-bulk decomposition of generalized Bethe states given in Proposition 3.2, we can now compute the action of local operators on these states similarly as in [1]. This action is however quite cumbersome due to the fact that we have to deal with generalized gauged transformed bulk and boundary operators. To overcome this problem, we identified in [1] a particular basis in the space of local operators on the first $m$ sites of the chain, for which the action on generalized Bethe states happens to have a relatively simple combinatorial form.[4] We recall here the form of this basis and compute the action of its elements on states of the form (3.24), hence generalizing the result of [1] to the case of a general boundary matrix $K_-$.

Let $E^{i,j}$, $i,j \in \{1,2\}$, denote the usual elementary matrices on $\mathbb{C}^2$, i.e. the $2\times 2$ matrices with elements $(E^{i,j})_{k,\ell}=\delta_{i,k}\delta_{j,\ell}$. As in [1], we consider the following local operators at site $n$:

$$E_n^{\epsilon'_n,\epsilon_n}(u|(a,b),(\bar{a},\bar{b})) = S_n(-u|\bar{a},\bar{b})\,E_n^{\epsilon'_n,\epsilon_n}\,[S_n(-u|a,b)]^{-1} \in \mathrm{End}\,\mathcal{H}_n\,,\qquad (4.1)$$

with $\epsilon'_n,\epsilon_n \in \{1,2\}$. For given arbitrary values of the parameters $(u,a,b,\bar{a},\bar{b})$, the operators (4.1) are therefore four different linear combinations of the local elementary operators $E_n^{i,j} \in \mathrm{End}\,\mathcal{H}_n$, $1 \leq i,j \leq 2$, with coefficients written in terms of $(u,a,b,\bar{a},\bar{b})$.[5] From (4.1), we

---

[4]This action is in fact similar to the action of the natural basis of local operators on the usual Bethe states that was obtained in the diagonal case [20].

[5]Explicitly, we have

$$E_n^{1,1}(u|(a,b),(\bar{a},\bar{b})) = \frac{e^{\eta a}}{2\sinh(\eta b)}\big[-e^{-\eta(\bar{a}+\bar{b})}E_n^{1,1}+e^{-u-\eta(a+\bar{a}-b+\bar{b})}E_n^{1,2}-e^u E_n^{2,1}+e^{-\eta(a-b)}E_n^{2,2}\big],$$

$$E_n^{1,2}(u|(a,b),(\bar{a},\bar{b})) = \frac{e^{\eta a}}{2\sinh(\eta b)}\big[e^{-\eta(\bar{a}+\bar{b})}E_n^{1,1}-e^{-u-\eta(a+\bar{a}+b+\bar{b})}E_n^{1,2}+e^u E_n^{2,1}-e^{-\eta(a+b)}E_n^{2,2}\big],$$

$$E_n^{2,1}(u|(a,b),(\bar{a},\bar{b})) = \frac{e^{\eta a}}{2\sinh(\eta b)}\big[-e^{-\eta(\bar{a}-\bar{b})}E_n^{1,1}+e^{-u-\eta(a+\bar{a}-b-\bar{b})}E_n^{1,2}-e^u E_n^{2,1}+e^{-\eta(a-b)}E_n^{2,2}\big],$$

$$E_n^{2,2}(u|(a,b),(\bar{a},\bar{b})) = \frac{e^{\eta a}}{2\sinh(\eta b)}\big[e^{-\eta(\bar{a}-\bar{b})}E_n^{1,1}-e^{-u-\eta(a+\bar{a}+b-\bar{b})}E_n^{1,2}+e^u E_n^{2,1}-e^{-\eta(a+b)}E_n^{2,2}\big].$$

define the following set of tensor products of such local operators on the first $m$ sites of the chain:

$$\mathbb{E}_m(\alpha,\beta) = \left\{ \prod_{n=1}^{m} E_n^{\epsilon'_n,\epsilon_n}(\xi_n|(a_n,b_n),(\bar{a}_n,\bar{b}_n)) \in \mathrm{End}(\otimes_{n=1}^{m}\mathcal{H}_n) \mid \epsilon,\epsilon' \in \{1,2\}^m \right\}, \quad (4.2)$$

in which the parameters $a_n, \bar{a}_n, b_n, \bar{b}_n, 1 \le n \le m$, are fixed in terms of the gauged parameters $\alpha, \beta$ and of the $m$-tuples $\epsilon \equiv (\epsilon_1,\dots,\epsilon_m)$ and $\epsilon' \equiv (\epsilon'_1,\dots,\epsilon'_m)$ as[6]

$$a_n = \alpha + 1, \qquad b_n = \beta - \sum_{r=1}^{n}(-1)^{\epsilon_r}, \tag{4.3}$$

$$\bar{a}_n = \alpha - 1, \qquad \bar{b}_n = \beta + \sum_{r=n+1}^{m}(-1)^{\epsilon'_r} - \sum_{r=1}^{m}(-1)^{\epsilon_r} = b_n + 2\tilde{m}_{n+1}, \tag{4.4}$$

with

$$\tilde{m}_n = \sum_{r=n}^{m}(\epsilon'_r - \epsilon_r) = \sum_{r=n}^{m}\frac{(-1)^{\epsilon'_r}-(-1)^{\epsilon_r}}{2}. \tag{4.5}$$

Then, we have shown in [1] that, except for a finite numbers of values of $\beta \bmod 2\pi/\eta$, the set $\mathbb{E}_m(\alpha,\beta)$ defines a basis of $\mathrm{End}(\otimes_{n=1}^{m}\mathcal{H}_n)$. In other words, it means that any operator $\mathcal{O}_{1\to m}$ which acts only on the first $m$ sites of the chain can be expressed as a linear combination of elements of $\mathbb{E}_m(\alpha,\beta)$.

The boundary bulk decomposition of boundary states given in Proposition 3.2, together with the action of the elements of this basis on gauged bulk Bethe states obtained in Theorem 4.1 of [1], enables us to compute the action of these elements on gauged boundary Bethe states of the form

$$\widehat{\underline{\mathcal{B}}}_{-,M}(\{\lambda_i\}_{i=1}^{M}|\alpha-\beta+1)\,|\,\eta,\alpha+\beta+N-2M-1\rangle \tag{4.6}$$

under the following constraint on the choice on the gauge parameters $\alpha$ and $\beta$:

$$\eta(\alpha+\beta+N-2M-1) = -\tau_+ - \epsilon_+(\varphi_+ + \psi_+) + \frac{\epsilon_+ - 1}{2}i\pi \quad \bmod 2i\pi, \tag{4.7}$$

with $\epsilon_+ = \pm 1$. More precisely, we can formulate the following result, which is the analog of Theorem 4.2 of [1] for this case with an arbitrary boundary matrix $K_-$, provided that we choose $\alpha$ and $\beta$ satisfying (4.7):[7]

**Theorem 4.1.** *Under any choice of $\alpha,\beta$ satisfying (4.7), the action on the state (4.6) of a generic element $\prod_{n=1}^{m} E_n^{\epsilon'_n,\epsilon_n}(\xi_n|(a_n,b_n),(\bar{a}_n,\bar{b}_n))$ of the basis (4.2) of local operators on the first $m$ sites of the chain, where the parameters $a_n, b_n, \bar{a}_n, \bar{b}_n$ are defined by (4.3) and (4.4), is given as*

$$\prod_{n=1}^{m} E_n^{\epsilon'_n,\epsilon_n}(\xi_n|(a_n,b_n),(\bar{a}_n,\bar{b}_n))\widehat{\underline{\mathcal{B}}}_{-,M}(\{\mu_i\}_{i=1}^{M}|\alpha-\beta+1)\,|\,\eta,\alpha+\beta+N-1-2M\rangle$$

$$= \sum_{\mathsf{B}_{\epsilon,\epsilon'}} \bar{\mathcal{F}}_{\mathsf{B}_{\epsilon,\epsilon'}}(\{\mu_j\}_{j=1}^{M},\{\xi_j^{(1)}\}_{j=1}^{m}|\beta)$$

$$\times \widehat{\underline{\mathcal{B}}}_{-,M+\tilde{m}_{\epsilon,\epsilon'}}(\{\mu_i\}_{\substack{i=1\\i\notin\mathsf{B}_{\epsilon,\epsilon'}}}^{M+m}|\alpha-\beta+1-2\tilde{m}_{\epsilon,\epsilon'})\,|\,\eta,\alpha+\beta+N-1-2M\rangle. \tag{4.8}$$

---

[6]For instance, for $m=1$, the $b$ and $\bar{b}$-parameters are fixed as $b_1 = \bar{b}_1 = \beta - (-1)^{\epsilon_1}$; for $m=2$, they are fixed as $b_1 = \beta - (-1)^{\epsilon_1}$, $\bar{b}_2 = b_2 = b_1 - (-1)^{\epsilon_2} = \beta - (-1)^{\epsilon_1} - (-1)^{\epsilon_2}$, and $\bar{b}_1 = b_2 + (-1)^{\epsilon'_2}$.

[7]Note that the formulation of this result does not impose any condition on $\alpha - \beta$.

Here, we have defined $\mu_{M+j} := \xi^{(1)}_{m+1-j}$ for $j \in \{1, \dots, m\}$. The sum in (4.8) runs over all possible sets of integers $B_{\epsilon,\epsilon'} = \{B_1, \dots, B_{s+s'}\}$ whose elements satisfy the conditions

$$
\begin{cases}
B_p \in \{1, \dots, M\} \setminus \{B_1, \dots, B_{p-1}\} & \text{for} \quad 0 < p \le s, \\
B_p \in \{1, \dots, M+m+1-i_p\} \setminus \{B_1, \dots, B_{p-1}\} & \text{for} \quad s < p \le s+s'.
\end{cases}
\tag{4.9}
$$

with the notations

$$
\{i_p\}_{p \in \{1, \dots, s\}} = \{1, \dots, m\} \cap \{j \mid \epsilon_j = 2\} \qquad \text{with} \quad i_p < i_q \ \text{if} \ p < q,
\tag{4.10}
$$

$$
\{i_p\}_{p \in \{s+1, \dots, s+s'\}} = \{1, \dots, m\} \cap \{j \mid \epsilon'_j = 1\} \qquad \text{with} \quad i_p > i_q \ \text{if} \ p < q.
\tag{4.11}
$$

Moreover,

$$
\begin{aligned}
\bar{\mathcal{F}}_{B_{\epsilon,\epsilon'}}(\{\mu_j\}_{j=1}^M, \{\xi^{(1)}_j\}_{j=1}^m \mid \beta) &= (-1)^{(N+1)\tilde{m}_{\epsilon,\epsilon'}} e^{\eta \tilde{m}_{\epsilon,\epsilon'}(\beta+\tilde{m}_{\epsilon,\epsilon'})} \prod_{n=1}^m \frac{e^\eta}{\sinh(\eta b_n)} \sum_{\sigma_{\alpha_+} = \pm} \frac{\prod_{j=1}^{s+s'} d(\mu^\sigma_{B_j})}{\prod_{j=1}^m d(\xi^{(1)}_j)} \\
&\times \frac{H_{\sigma_{\alpha_+}}(\{\mu_{\alpha_+}\})}{H_1(\{\xi^{(1)}_{\gamma_+}\})} \prod_{i \in \alpha_-} \prod_{\epsilon = \pm} \left\{ \prod_{j \in \alpha_+} \frac{\sinh(\mu^\sigma_j + \epsilon\mu_i + \eta)}{\sinh(\mu^\sigma_j + \epsilon\mu_i)} \prod_{j \in \gamma_+} \frac{\sinh(\xi^{(1)}_j + \epsilon\mu_i)}{\sinh(\xi^{(0)}_j + \epsilon\mu_i)} \right\} \\
&\times \prod_{i \in \alpha_+} \left\{ \prod_{j \in \gamma_+} \frac{\sinh(\xi^{(1)}_j - \mu^\sigma_i)}{\sinh(\xi^{(0)}_j - \mu^\sigma_i)} \frac{\prod_{j \in \alpha_+} \sinh(\mu^\sigma_j - \mu^\sigma_i - \eta)}{\prod_{j \in \alpha_+ \setminus \{i\}} \sinh(\mu^\sigma_j - \mu^\sigma_i)} \right\} \prod_{1 \le i < j \le s+s'} \frac{\sinh(\mu^\sigma_{B_i} - \mu^\sigma_{B_j})}{\sinh(\mu^\sigma_{B_i} - \mu^\sigma_{B_j} - \eta)} \\
&\times \prod_{p=1}^s \left[ \sinh(\xi^{(1)}_{i_p} - \mu^\sigma_{B_p} + \eta(1+b_{i_p})) \frac{\prod_{k=i_p+1}^m \sinh(\mu^\sigma_{B_p} - \xi^{(1)}_k - \eta)}{\prod_{k=i_p}^m \sinh(\mu^\sigma_{B_p} - \xi^{(1)}_k)} \right] \\
&\times \prod_{p=s+1}^{s+s'} \left[ \sinh(\xi^{(1)}_{i_p} - \mu^\sigma_{B_p} - \eta(1-\bar{b}_{i_p})) \frac{\prod_{k=i_p+1}^m \sinh(\xi^{(1)}_k - \mu^\sigma_{B_p} - \eta)}{\prod_{\substack{k=i_p \\ k \ne M+m+1-B_p}}^m \sinh(\xi^{(1)}_k - \mu^\sigma_{B_p})} \right],
\end{aligned}
\tag{4.12}
$$

where the sum is performed over all $\sigma_j \in \{+, -\}$ for $j \in \alpha_+$, we have defined $\mu^\sigma_i = \sigma_i \mu_i$ for $i \in B_{\epsilon,\epsilon'}$, with $\sigma_i = 1$ if $i > M$, and

$$
\alpha_+ = B_{\epsilon,\epsilon'} \cap \{1, \dots, M\}, \qquad\qquad \alpha_- = \{1, \dots, M\} \setminus \alpha_+,
\tag{4.13}
$$

$$
\gamma_- = \{M+m+1-j\}_{j \in B_{\epsilon,\epsilon'} \cap \{N+1, \dots, N+m\}}, \quad \gamma_+ = \{1, \dots, m\} \setminus \gamma_-.
\tag{4.14}
$$

The function $H_\sigma(\{\lambda\}) \equiv H_\sigma(\{\lambda\} \mid \epsilon_+ \varphi_+, \epsilon_+ \psi_+)$ is given by (3.25).

*Proof.* Noticing that the constraint (4.7) is compatible with the choice (3.13)-(3.14) of $\gamma$ and $\delta$ such that

$$
\alpha + \beta + N - 2M - 1 = \gamma + \delta,
\tag{4.15}
$$

we can use the boundary-bulk decomposition of Proposition 3.2 to decompose the gauge boundary Bethe state (4.6) into gauged bulk Bethe state of the form

$$
B(\lambda^{(\sigma)}_1 \mid x-1, \alpha-\beta) \cdots B(\lambda^{(\sigma)}_M \mid x-M, \alpha-\beta+M-1) \mid \eta, \alpha+\beta+N-M-1 \rangle
\tag{4.16}
$$

with $x = \gamma - \delta$. We then use Theorem 4.1 of [1] to compute the action of $\prod_{n=1}^m E^{\epsilon'_n, \epsilon_n}_n(\xi_n \mid (a_n, b_n), (\bar{a}_n, \bar{b}_n))$ on the bulk states (4.16) and use again Proposition 3.2 to reconstruct boundary states. By doing this, we get rid of the apparent dependance on $\gamma - \delta$.
□

*Remark* 2. The action presented in Theorem 4.1 formally coincides with the action that we have computed in Theorem 4.2 of [1]. However, one has to remark that there it was derived under the very strong boundary condition:

$$K_{-,0}(\lambda;\varsigma_+ = -\infty,\kappa_+,\tau_+) = \begin{pmatrix} e^{(\eta/2-\lambda)} & 0 \\ 0 & e^{(\lambda-\eta/2)} \end{pmatrix}_0 = e^{(\eta/2-\lambda)\sigma_0^z}, \qquad (4.17)$$

which completely fixes the boundary field at site $N$. Here, instead, we are leaving the three boundary parameters $\varsigma_+$, $\kappa_+$, $\tau_+$ completely arbitrary. The price to pay is that the gauge parameters $\alpha$ and $\beta$ need to satisfy the constraint (4.7), whereas in [1] they could be taken arbitrary. It happens that the boundary-bulk decomposition here derived in Proposition 3.2 shows that, even under general boundary parameters, the gauged transformed boundary Bethe states admit formally, under special choice of the gauge parameters $\gamma$ and $\delta$, the same boundary-bulk decomposition as in the special boundary case (4.17), just the boundary-bulk coefficients $H_\sigma(\{\lambda\})$ here account for the dependence from the current general values of the boundary parameters $\varsigma_+$, $\kappa_+$, $\tau_+$.

We recall that, under the constraint (2.42) on the boundary parameters, any separate state in the Sklyanin's SoV approach can be expressed as a generalized Bethe state of the form (4.6) with the choice (2.43)-(2.44) of $\alpha$ and $\beta$, see Proposition 2.2. Note that the choice (2.43)-(2.44) together with the constraint (2.42) imposes that $\alpha$ and $\beta$ satisfy the constraint (4.7) with $\epsilon_+ = \epsilon_{\varphi_+}$:

$$\eta(\alpha+\beta) = -\tau_- + \epsilon_{\varphi_-}(\varphi_- - \psi_-) + \frac{1-\epsilon_{\varphi_-}}{2}i\pi \mod 2i\pi,$$

$$= -\tau_+ - \epsilon_{\varphi_+}(\varphi_+ + \psi_+) - (N-1-2M)\eta + \frac{1-\epsilon_{\varphi_+}}{2}i\pi \mod 2i\pi. \qquad (4.18)$$

Hence Theorem 4.1 also provides the action of a generic element $\prod_{n=1}^m E_n^{\epsilon'_n,\epsilon_n}(\xi_n|(a_n,b_n),(\bar{a}_n,\bar{b}_n))$ of the basis (4.2) on any generic Sklyanin's separate state, provided that the constraint (2.42) is satisfied. We see that the result can still be expressed as a linear combination of separate states if $\tilde{m}_{\epsilon,\epsilon'} = 0$.

# 5 Matrix elements of $m$-site local operator strips in the finite chain

We now consider the matrix elements of an element of the basis (4.2) of strips of local operators on the first $m$-sites of the chain, in a generic transfer matrix eigenstate associated with a solution of homogeneous Bethe equations under the non-diagonal boundary conditions subject to the constraint (2.42), that we recall here:

$$\tau_+ - \tau_- = -\epsilon_{\varphi_+}(\varphi_+ + \psi_+) - \epsilon_{\varphi_-}(\varphi_- - \psi_-) - (N-1-2M)\eta + \frac{1-\epsilon_{\varphi_+}\epsilon_{\varphi_-}}{2}i\pi \mod 2i\pi \quad (5.1)$$

for a given $M \in \{1,\dots,N\}$ and with a given choice of $\epsilon_{\varphi_+},\epsilon_{\psi_+} \in \{+1,-1\}$. More precisely, we consider a matrix element of the form

$$\langle\{\lambda\}| \prod_{n=1}^m E_n^{\epsilon'_n,\epsilon_n}(\xi_n|(a_n,b_n),(\bar{a}_n,\bar{b}_n))|\{\lambda\}\rangle = \frac{\langle Q| \prod_{n=1}^m E_n^{\epsilon'_n,\epsilon_n}(\xi_n|(a_n,b_n),(\bar{a}_n,\bar{b}_n))|Q\rangle}{\langle Q|Q\rangle}, \quad (5.2)$$

in which $|\{\lambda\}\rangle$ and $\langle\{\lambda\}|$ are the normalized transfer matrix eigenstates associated with a solution $\{\lambda\} \equiv \{\lambda_1,\dots,\lambda_M\}$ of the homogeneous Bethe equations under the constraint (5.1). We denote by $Q$ the associated polynomial of degree $M$ of the form (2.16) with roots given by

$\{\lambda\}$, solution of the homogeneous $TQ$-equation (2.38) with the corresponding transfer matrix eigenvalue $\tau$. Hence, in the notations (5.2), $\langle Q|$ and $|Q\rangle$ denote the SoV eigenstates constructed in the Sklyanin's approach.[8] We fix here $\alpha$ and $\beta$ as in (2.43)-(2.44), and $a_n, b_n, \bar{a}_n, \bar{b}_n$ are given in terms of $\alpha, \beta, \epsilon, \epsilon'$ as in (4.3)-(4.4).

It follows from Proposition 2.2 that (5.2) can be re-expressed as

$$
\langle\{\lambda\}|\prod_{n=1}^{m} E_n^{\epsilon'_n,\epsilon_n}(\xi_n|(a_n,b_n),(\bar{a}_n,\bar{b}_n))|\{\lambda\}\rangle = c_{M,\mathrm{ref}}^{(R)}
$$
$$
\times \frac{\langle Q|\prod_{n=1}^{m} E_n^{\epsilon'_n,\epsilon_n}(\xi_n|(a_n,b_n),(\bar{a}_n,\bar{b}_n))\widehat{\underline{\mathcal{B}}}_{-,M}(\{\lambda_i\}_{i=1}^{M}|\alpha-\beta+1)|\eta,\alpha+\beta+N-2M-1\rangle}{\langle Q|Q\rangle},
$$
$$(5.3)$$

and we can now use Theorem 4.1 to act with $\prod_{n=1}^{m} E_n^{\epsilon'_n,\epsilon_n}(\xi_n|(a_n,b_n),(\bar{a}_n,\bar{b}_n))$ on the right in this expression. We restrict here our study to the case in which[9]

$$
\sum_{r=1}^{m}(\epsilon'_r-\epsilon_r)=0,
$$
$$(5.4)$$

for which the result of this action can again be directly expressed as a sum over Sklyanin's separate states of a similar type:

$$
\langle\{\lambda\}|\prod_{n=1}^{m} E_n^{\epsilon'_n,\epsilon_n}(\xi_n|(a_n,b_n),(\bar{a}_n,\bar{b}_n))|\{\lambda\}\rangle = \sum_{\mathsf{B}_{\epsilon,\epsilon'}} \bar{\mathcal{F}}_{\mathsf{B}_{\epsilon,\epsilon'}}(\{\lambda\},\{\xi_j^{(1)}\}_{j=1}^{m}|\beta)\frac{\langle Q|\bar{Q}_{\mathsf{B}_{\epsilon,\epsilon'}}\rangle}{\langle Q|Q\rangle}. \quad (5.5)
$$

Here we have used the notations of Theorem 4.1, and $\bar{Q}_{\mathsf{B}_{\epsilon,\epsilon'}}$ denotes the polynomial of the form (2.16) with roots given by

$$
\{\bar{\lambda}_1,\ldots,\bar{\lambda}_M\} \equiv \{\lambda_1,\ldots,\lambda_{N+m}\} \setminus \{\lambda_j\}_{j\in\mathsf{B}_{\epsilon,\epsilon'}} = \{\lambda_a\}_{a\in\alpha_-} \cup \{\xi_{i_b}^{(1)}\}_{b\in\alpha_+}, \quad (5.6)
$$

in which, as in Theorem 4.1, we have denoted $\xi_{m+1-j}^{(1)} \equiv \lambda_{M+j}$ for $j\in\{1,\ldots,m\}$. We can then use the formulas obtained in [40] to compute the resulting scalar products of separate states.

The technical details of the computation are then completely similar to those performed in [1], and we therefore obtain the following result, which extends to the case (5.1) the result stated in Theorem 5.1 of [1]:

**Theorem 5.1.** *Let the boundary condition* (5.1), *for a given* $M\in\{1,\ldots,N\}$ *and with a given choice of* $\epsilon_{\varphi_+},\epsilon_{\psi_+}\in\{+1,-1\}$, *be satisfied, and let* $\{\lambda\}\equiv\{\lambda_1,\ldots,\lambda_M\}$ *be a solution of the homogeneous Bethe equations under the constraint* (5.1). *Let* $\alpha$ *and* $\beta$ *be given in terms of* $\varphi_-,\psi_-$ *by* (2.43)-(2.44), *and let* $\epsilon\equiv(\epsilon_1,\ldots,\epsilon_m),\epsilon'\equiv(\epsilon'_1,\ldots,\epsilon'_m)\in\{1,2\}^m$ *satisfying* (5.4).

---

[8]In the expression (5.2), $\langle Q|$ and $|Q\rangle$ may as well denote the SoV eigenstates constructed in the new approach of [80] since they are proportional to the Sklyanin's ones due to the simplicity of the transfer matrix spectrum. However, when acting on the generalized Bethe state in (5.3), we obtain a sum over generalized Bethe states with modified arguments which are no longer transfer matrix eigenstates, see Theorem 4.1. Under the condition (5.4), these generalized Bethe states can be re-expressed as Sklyanin's separate states thanks to Proposition 2.2, but a priori not in terms of the type of separate states obtained within the new approach of [80]. Therefore our computations need in principle to be performed in the generalized Sklyanin's SoV framework. Note however that, if we happen to be in a limiting case in which only the new separation of variables approach is applicable, we can proceed as in [1] by taking limits from regions of the space of parameters in which Sklyanin's SoV is applicable.

[9]For matrix elements for which (5.4) is not satisfied, we encounter the same problem as in [1]: the action of Theorem 4.1 produces states with shifted gauge parameters and shifted number of roots, that at the moment we do not know how to re-express in a simple form in terms of usual types of separate states.

*Then the matrix elements* (5.2), *where* $a_n, b_n, \bar{a}_n, \bar{b}_n$ *are given in terms of* $\alpha, \beta, \epsilon, \epsilon'$ *as in* (4.3)-(4.4), *can be written as*

$$\langle\{\lambda\}|\prod_{n=1}^{m} E_n^{\epsilon'_n,\epsilon_n}(\xi_n|(a_n,b_n),(\bar{a}_n,\bar{b}_n))|\{\lambda\}\rangle$$
$$= \sum_{\text{B}_1=1}^{M}\cdots\sum_{\text{B}_s=1}^{M}\sum_{\text{B}_{s+1}=1}^{M+m}\cdots\sum_{\text{B}_m=1}^{M+m}\frac{H_{\{\text{B}_j\}}(\{\lambda\}|\beta)}{\prod_{1\le l<k\le m}\sinh(\xi_k-\xi_l)\sinh(\xi_k+\xi_l)}, \quad (5.7)$$

*with*

$$H_{\{\text{B}_j\}}(\{\lambda\}|\beta) = \prod_{n=1}^{m}\frac{e^{\eta}}{\sinh(\eta b_n)}\sum_{\sigma_{\text{B}_j}}\frac{(-1)^s\prod_{i=1}^{m}\sigma_{\text{B}_i}\prod_{i=1}^{m}\prod_{j=1}^{m}\sinh(\lambda_{\text{B}_i}^{\sigma}+\xi_j+\eta/2)}{\prod_{1\le i<j\le m}\sinh(\lambda_{\text{B}_i}^{\sigma}-\lambda_{\text{B}_j}^{\sigma}-\eta)\sinh(\lambda_{\text{B}_i}^{\sigma}+\lambda_{\text{B}_j}^{\sigma}+\eta)}$$

$$\times\prod_{p=1}^{s}\left\{\sinh(\lambda_{\text{B}_p}^{\sigma}-\xi_{i_p}^{(1)}-\eta(1+b_{i_p}))\prod_{k=1}^{i_p-1}\sinh(\lambda_{\text{B}_p}^{\sigma}-\xi_k^{(1)})\prod_{k=i_p+1}^{m}\sinh(\lambda_{\text{B}_p}^{\sigma}-\xi_k^{(0)})\right\}$$

$$\times\prod_{p=s+1}^{m}\left\{\sinh(\lambda_{\text{B}_p}^{\sigma}-\xi_{i_p}^{(1)}+\eta(1-\bar{b}_{i_p}))\prod_{k=1}^{i_p-1}\sinh(\lambda_{\text{B}_p}^{\sigma}-\xi_k^{(1)})\prod_{k=i_p+1}^{m}\sinh(\lambda_{\text{B}_p}^{\sigma}-\xi_k^{(1)}+\eta)\right\}$$

$$\times\prod_{k=1}^{m}\frac{\sinh(\xi_k-\epsilon_{\varphi_-}\varphi_-)\cosh(\xi_k-\epsilon_{\varphi_-}\psi_-)}{\sinh(\lambda_{\text{B}_k}^{\sigma}-\epsilon_{\varphi_-}\varphi_-+\eta/2)\cosh(\lambda_{\text{B}_k}^{\sigma}-\epsilon_{\varphi_-}\psi_-+\eta/2)}\det_m\Omega. \quad (5.8)$$

*Here we have used the notations* (4.10)-(4.11), *and the sum is performed over all* $\sigma_{\text{B}_j}\in\{+,-\}$
*for* $\text{B}_j\le M$, *and* $\sigma_{\text{B}_j}=1$ *for* $\text{B}_j>M$. *Finally, the* $m\times m$ *matrix* $\Omega$ *is given by*

$$\Omega_{lk}=-\delta_{N+m+1-b_l,k}, \qquad \text{for } \text{B}_l>M, \quad (5.9)$$

$$\Omega_{lk}=\sum_{a=1}^{M}\left[\mathcal{N}^{-1}\right]_{\text{B}_l,a}\mathcal{M}_{a,k}, \quad \text{for } \text{B}_l\le M, \quad (5.10)$$

*where we have defined*

$$[\mathcal{N}]_{j,k}\equiv[\mathcal{N}(\boldsymbol{\lambda})]_{j,k}=2N\,\delta_{j,k}\,\Xi'_{\boldsymbol{\epsilon},Q}(\lambda_j)+2\pi\left[K(\lambda_j-\lambda_k)-K(\lambda_j+\lambda_k)\right], \quad (5.11)$$

$$[\mathcal{M}]_{j,k}\equiv[\mathcal{M}(\bar{\boldsymbol{\lambda}},\boldsymbol{\lambda})]_{j,k}=\begin{cases}[\mathcal{N}(\boldsymbol{\lambda})]_{j,k} & \text{if } k\in\alpha_-,\\ i[t(\xi_{i_k}-\lambda_j)-t(\xi_{i_k}+\lambda_j)] & \text{if } k\in\alpha_+,\end{cases} \quad (5.12)$$

*in terms of*

$$\Xi'_{\boldsymbol{\epsilon},Q}(\mu)=\frac{i}{2N}\frac{\partial}{\partial\mu}\left(\log\frac{\mathbf{A}_{\boldsymbol{\epsilon}}(-\mu)Q(\mu+\eta)}{\mathbf{A}_{\boldsymbol{\epsilon}}(\mu)Q(\mu-\eta)}\right), \quad (5.13)$$

$$K(\lambda)=\frac{i\sinh(2\eta)}{2\pi\sinh(\lambda+\eta)\sinh(\lambda-\eta)}=\frac{i}{2\pi}\left[t(\lambda+\eta/2)+t(\lambda-\eta/2)\right], \quad (5.14)$$

$$t(\lambda)=\frac{\sinh\eta}{\sinh(\lambda-\eta/2)\sinh(\lambda+\eta/2)}=\coth(\lambda-\eta/2)-\coth(\lambda+\eta/2). \quad (5.15)$$

## 6 On correlation functions in the half-infinite chain

As mentioned above, the description of the transfer matrix spectrum and eigenstates in terms of the homogeneous *TQ*-equation (2.38) for *Q* of the form (2.16) is not complete under the

constraint (5.1). However, it was argued in [13] that, at least for the range of boundary parameters numerically investigated there, the corresponding Bethe equations,

$$\mathbf{A}_{\boldsymbol{\varepsilon}}(\lambda_j)Q(\lambda_j - \eta) + \mathbf{A}_{\boldsymbol{\varepsilon}}(-\lambda_j)Q(\lambda_j + \eta) = 0, \qquad j = 1, \ldots, M, \tag{6.1}$$

which can be rewritten as, for $\lambda_j \neq 0, i\frac{\pi}{2}$,

$$\frac{a(-\lambda_j)d(\lambda_j)}{a(\lambda_j)d(-\lambda_j)} \prod_{\sigma=\pm} \frac{\sinh(\lambda_j + \frac{\eta}{2} - \epsilon_{\varphi_\sigma}\varphi_\sigma)\cosh(\lambda_j + \frac{\eta}{2} - \sigma\epsilon_{\varphi_\sigma}\psi_\sigma)}{\sinh(\lambda_j - \frac{\eta}{2} + \epsilon_{\varphi_\sigma}\varphi_\sigma)\cosh(\lambda_j - \frac{\eta}{2} + \sigma\epsilon_{\varphi_\sigma}\psi_\sigma)}$$

$$\times \prod_{\substack{k=1\\k\neq j}}^{M} \frac{\sinh(\lambda_j - \lambda_k + \eta)\sinh(\lambda_j + \lambda_k + \eta)}{\sinh(\lambda_j - \lambda_k - \eta)\sinh(\lambda_j + \lambda_k - \eta)} = 1, \qquad j = 1, \ldots, M, \tag{6.2}$$

yield the ground state in the sector

$$M = \left\lfloor \frac{N}{2} \right\rfloor. \tag{6.3}$$

It was more generally conjectured in [14] that, when combining the solutions in the sector $M$ with the constraint (5.1) with a given choice of $\epsilon_{\varphi_+}, \epsilon_{\varphi_-}$ and the solutions in the sector $M' = N - 1 - M$ with a constraint in which the signs are negated, one gets the complete spectrum. If this conjecture is true, this means in particular that, under the constraint (5.1), the ground state can be found in one the two sectors $(M, \boldsymbol{\varepsilon})$ or $(M', -\boldsymbol{\varepsilon})$, and can be described by usual Bethe equations. Note that, if $N - 2M$ remains finite in the thermodynamic limit, so is $N - 2M'$, and we can consider the half-infinite chain limit (thermodynamic limit $N \to \infty$) while maintaining the constraint (5.1).

If we are in such a situation, then the analysis concerning the distribution of Bethe roots in the ground state can be performed completely similarly as in the diagonal case [8, 9, 41]. Hence, the ground state should be characterized in the homogeneous and thermodynamic limits by a infinite number (i.e. of order $N/2$) of Bethe roots distributed on an interval $(0, \Lambda)$ according to a density function $\rho(\lambda)$, with

$$\rho(\lambda) = \begin{cases} \dfrac{1}{\zeta\cosh(\pi\lambda/\zeta)} & \text{with } \zeta = i\eta > 0 \qquad \text{and } \Lambda = +\infty \qquad \text{if } |\Delta| < 1, \\[2ex] \dfrac{i}{\pi}\dfrac{\vartheta'_1(0,q)}{\vartheta_2(0,q)}\dfrac{\vartheta_3(i\lambda,q)}{\vartheta_4(i\lambda,q)} & \text{with } q = e^\eta \ (\eta < 0) \quad \text{and } \Lambda = -i\pi/2 \quad \text{if } \Delta > 1. \end{cases} \tag{6.4}$$

This density function is solution of the following integral equation:

$$\rho(\lambda) + \int_0^\Lambda \left[K(\lambda - \mu) + K(\lambda + \mu)\right]\rho(\mu)\,d\mu = \frac{it(\lambda)}{\pi}. \tag{6.5}$$

which can be extended by parity on the whole interval $(-\Lambda, \Lambda)$ as

$$\rho(\lambda) + \int_{-\Lambda}^\Lambda K(\lambda - \mu)\rho(\mu)\,d\mu = \frac{it(\lambda)}{\pi}. \tag{6.6}$$

Depending on the configuration of the boundary parameters and on the precise number of Bethe roots $\lfloor\frac{N}{2}\rfloor - k$ (with $k$ remaining finite in the thermodynamic limit) characterising the ground state, the fine structure of the ground state may also include a finite number of "complex" roots,[10] i.e. of roots that are not on the real axis in the regime $|\Delta| < 1$ nor on the

---

[10]We may also possibly have a finite number of "holes" in the distribution of "real" roots. The latter should however not contribute to the leading order of the result for the correlation functions in the thermodynamic limit.

imaginary axis in the regime $\Delta > 1$. In particular, it may contain some isolated "complex" roots of the form

$$\check{\lambda}_{\sigma,1} = \eta/2 - \epsilon_{\varphi_\sigma}\varphi_\sigma + \varepsilon_{\sigma,1}, \qquad \check{\lambda}_{\sigma,2} = \eta/2 - \sigma\epsilon_{\varphi_\sigma}\psi_\sigma + i\frac{\pi}{2} + \varepsilon_{\sigma,2}, \qquad \sigma = +,-, \quad (6.7)$$

which converge towards the poles of the boundary factor in (6.2) with exponentially small corrections $\varepsilon_{\sigma,i}$ in $N$: these roots play an important role in the computation of the correlation functions in the thermodynamic limit (see [20,41]), and we call them boundary roots.

Although these are interesting problems, we shall not discuss here the validity of the aforementioned conjectures, nor the fine structure of the ground state according to the precise configuration of the boundary fields. Instead, we shall simply assume that we are in a configuration of the boundaries so that the ground state can be characterized, in the thermodynamic limit, by an infinite set of Bethe roots on $(0, \Lambda)$ distributed according to the density function (6.4) solution of (6.5)-(6.6), with possibly some additional boundary roots of the form (6.7). Note that, although all types of boundary roots (6.7) may be present in the description of the ground state, only the presence of the boundary roots $\check{\lambda}_{-,1}, \check{\lambda}_{-,2}$ results into a non-zero direct contribution to the final result for the correlation functions around site 1 in the thermodynamic limit,[11] due to the presence of the singularities in the expression (5.8). Hence, we have here a priori to distinguish among the four following possible regimes:[12]

A) the set of Bethe roots for the ground state does not contain neither $\check{\lambda}_{-,1}$ nor $\check{\lambda}_{-,2}$;

B) the set of Bethe roots for the ground state contains $\check{\lambda}_{-,1}$ but not $\check{\lambda}_{-,2}$;

C) the set of Bethe roots for the ground state contains $\check{\lambda}_{-,2}$ but not $\check{\lambda}_{-,1}$;

D) the set of Bethe roots for the ground state contains both $\check{\lambda}_{-,1}$ and $\check{\lambda}_{-,2}$.

Under such assumptions, we can proceed as in [1,20] to compute the thermodynamic limit of the expression (5.7)-(5.8) for the ground state, and we obtain the following result for the mean value in the ground state of an element of the basis (4.2) under the condition (5.4) and under the hypothesis of Theorem 5.1:

$$\langle\prod_{n=1}^{m} E_n^{\epsilon'_n,\epsilon_n}(\xi_n|(a_n,b_n),(\bar{a}_n,\bar{b}_n))\rangle = \prod_{n=1}^{m}\frac{e^\eta}{\sinh(\eta b_n)}\frac{(-1)^s}{\prod_{j<i}\sinh(\xi_i-\xi_j)\prod_{i\le j}\sinh(\xi_i+\xi_j)}$$

$$\times \int_{\mathcal{C}}\prod_{j=1}^{s}d\lambda_j\int_{\mathcal{C}_\xi}\prod_{j=s+1}^{m}d\lambda_j\,H_m(\{\lambda_j\}_{j=1}^M;\{\xi_k\}_{k=1}^m)\det_{1\le j,k\le m}\left[\Phi(\lambda_j,\xi_k)\right], \quad (6.8)$$

---

[11]However, the fact that the set of Bethe roots for the ground state contains the boundary roots $\check{\lambda}_{-,1}$ and/or $\check{\lambda}_{-,2}$ may also depend on the boundary parameters at site $N$, see [41].

[12]For instance, the numerical results of Tables 2 and 4 of [13] seem to indicate that the numerical values of the boundary parameters considered in that paper correspond to regime A).

with

$$H_m(\{\lambda_j\}_{j=1}^M; \{\xi_k\}_{k=1}^m) = \frac{\prod_{j=1}^m \prod_{k=1}^m \sinh(\lambda_j + \xi_k + \eta/2)}{\prod_{1 \le i < j \le m} \sinh(\lambda_i - \lambda_j - \eta)\sinh(\lambda_i + \lambda_j + \eta)}$$

$$\times \prod_{p=1}^s \left\{ \sinh(\lambda_p - \xi_{i_p}^{(1)} - \eta(1 + b_{i_p})) \prod_{k=1}^{i_p-1} \sin(\lambda_p - \xi_k^{(1)}) \prod_{k=i_p+1}^m \sinh(\lambda_p - \xi_k^{(1)} - \eta) \right\}$$

$$\times \prod_{p=s+1}^m \left\{ \sinh(\lambda_p - \xi_{i_p}^{(1)} + \eta(1 - \bar{b}_{i_p})) \prod_{k=1}^{i_p-1} \sinh(\lambda_p - \xi_k^{(1)}) \prod_{k=i_p+1}^m \sinh(\lambda_p - \xi_k^{(1)} + \eta) \right\}$$

$$\times \prod_{k=1}^m \frac{\sinh(\xi_k - \epsilon_{\varphi_-}\varphi_-)\cosh(\xi_k - \epsilon_{\varphi_-}\psi_-)}{\sinh(\lambda_{B_k}^\sigma - \epsilon_{\varphi_-}\varphi_- + \eta/2)\cosh(\lambda_{B_k}^\sigma - \epsilon_{\varphi_-}\psi_- + \eta/2)}, \quad (6.9)$$

and

$$\Phi(\lambda_j, \xi_k) = \frac{1}{2}\left[\rho(\lambda_j - \xi_k) - \rho(\lambda_j + \xi_k)\right]. \quad (6.10)$$

The integration contours are defined as

$$\mathcal{C} = \begin{cases} [-\Lambda, \Lambda] & \text{in the regime A)}, \\ [-\Lambda, \Lambda] \cup \Gamma(\frac{\eta}{2} - \epsilon_{\varphi_-}\varphi_-) & \text{in the regime B)}, \\ [-\Lambda, \Lambda] \cup \Gamma(\frac{\eta}{2} + \epsilon_{\varphi_-}\psi_- + i\frac{\pi}{2}) & \text{in the regime C)}, \\ [-\Lambda, \Lambda] \cup \Gamma(\frac{\eta}{2} - \epsilon_{\varphi_-}\varphi_-, \frac{\eta}{2} + \epsilon_{\varphi_-}\psi_- + i\frac{\pi}{2}) & \text{in the regime D)}, \end{cases} \quad (6.11)$$

and

$$\mathcal{C}_\xi = \mathcal{C} \cup \Gamma(\xi_1^{(1)}, \ldots, \xi_m^{(1)}), \quad (6.12)$$

where $\Gamma(\mu_1, \ldots, \mu_p)$ denotes a contour surrounding the points of $\mu_1, \ldots, \mu_p$ with index 1, all other poles of the integrand being outside.

## 7 Conclusion and Outlooks

In this paper we have considered the open XXZ spin 1/2 chain with non-longitudinal boundary fields under one constraint relating the six boundary parameters. Under such a constraint, part of the spectrum and eigenstates of the transfer matrix can be described by a homogeneous $TQ$-equation, i.e. by usual Bethe equations [2]. The transfer matrix eigenstates can be constructed by Separation of Variables as particular separate states whose scalar product formulas are known [40]. Moreover, under the constraint, the separate states can be reformulated as generalized Bethe states which are written in terms of a gauged transformed reflection algebra obtained from the usual reflection algebra by a Vertex-IRF transformation.

So as to compute correlation functions, we have here derived the action of a basis of local operators on these generalized Bethe states under the most general boundary conditions on the site $N$. This gives us, in particular, the action of these local operators on the separate states, and therefore on the transfer matrix eigenstates which can be described by usual Bethe equations under the constraint. From this result and from the knowledge of the scalar product formula between separate states, we were able to derive multiple sum representations for the matrix elements, in these transfer matrix eigenstates, of the class of local operators which conserve the number of gauged $B$-operators under the action, i.e. which satisfy the constraint (5.4). Assuming that the ground state is among the eigenstates which can be described by usual Bethe

equations in a sector close to half-filling, as conjectured in [13], and that its corresponding set of Bethe roots is given in the thermodynamic by the usual density function with possibly extra boundary roots, we obtained multiple integral representation for the correlation functions of the aforementioned class of local operators. Our result generalizes those obtained in [1,20] for more restricted boundary conditions.

Several interesting questions remain however open.

As in [1], we had here to restrict our study to the class of local operators which satisfy the constraint (5.4). If this constraint is not satisfied, the action of the corresponding local operator on generalized Bethe states is expressed in terms of generalized Bethe states with a different numbers of $B$-operators and a shifted gauge parameter, see (4.8). The problem is that Proposition 2.2 does not enable us to re-express such generalized Bethe states in a simple way in terms of separate states. In other words, we do not know how to compute the resulting scalar products in a compact way after the action. It would be highly desirable, for the consideration of interesting physical correlation functions, to be able to understand how to overcome this difficulty.

Another interesting question concerns the validity of the conjecture of [13, 14], on which we relied to formulate our results. An analytical proof of this conjecture is still missing, so that the question remains whether it indeed holds for all ranges of boundary parameters satisfying the constraint. Also, an interesting question would be to identify precisely the fine structure of the ground state for the different values of the four independent boundary parameters appearing in the Bethe equations: identify in particular the boundary roots that are involved in its description, and discuss the resulting phase diagram. Such a proper and precise analysis of the ground state can a priori be done as in [41] for the diagonal case, but it is more involved since there are much more different cases to distinguish.

Finally, the question of an analytical computation of the correlation functions for completely generic boundary fields, i.e. not satisfying the constraint (2.42), still remains widely open. The difficulty comes from the actual description of the spectrum in terms of an inhomogeneous $TQ$-equation as in (2.36), which does not lead to a simple description of the ground state in terms of well-organized Bethe roots as in the homogeneous case (2.38). To overcome this problem, it would be necessary either to understand how to deal with such kind of equations within the analytical framework of the correlation functions, or to find an alternative, more convenient, description of the spectrum, maybe in terms of a homogenous $TQ$-equation with non-polynomial $Q$-solutions.

## Acknowledgments

**Funding information** G. N. is supported by CNRS and Laboratoire de Physique, ENS-Lyon. V. T. is supported by CNRS.

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
