# Peer review of "On correlation functions for the open XXZ chain with non-longitudinal boundary fields : the case with a constraint"

_SciPost Physics, doi:SciPost Phys. 16, 099 (2024)_

## Round 1 · Referee Report · Anonymous (Referee 1) · 2023-2-28

Strengths

  1. New results for the correlation functions of open spin chains
  2. Use of boundary-bulk decomposition for the gauged double-row monodromy matrix
  3. Multiple integral representations in the thermodynamic limit

Weaknesses

  1. Complicated form for the local operators due to the gauge transformation
  2. Relation between the constraint and selection of the ground state in the thermodynamic limit should be clarified

Report

The paper ``On correlation functions for the open XXZ chain with non-longitudinal boundary fields : the case with a constraint'' is a logical continuation of the work started by the same authors in their previous article. They study an extremely important case of the open spin chains with non-parallel boundary fields: the case with a boundary constraint relating the parameters on the sides of the lattice (compatible with usual homogeneous Baxter equation). The authors remind the construction of the eigenstates using the SOV approach to obtain finally the eigenstates in a form similar to the usual algebraic Bethe Ansatz. This representation in turn permits them to apply the boundary-bulk decomposition for the gauged operators and compute elements of the reduced density matrix as multiple sums and finally as multiple integrals in the thermodynamic limit. This result is new, important and extremely interesting for the study of open spin chains, for these reasons I recommend this paper for publication in SciPost Physics.

Requested changes

  1. The quantum inverse problem is solved in terms of gauge transformed local operators. The authors prove that these operators form a basis in the corresponding matrix space and thus give all the elements of the reduced density matrix. It would be very helpful to have some illustrations (for one or maybe two sites) of the local operators obtained by this procedure.

  2. The authors claim that the expression in thermodynamic limit works when the constraint fixes the number of Bethe roots at $N/2-k$ with $k$ remaining finite in the thermodynamic limit.The ground state density computed from the integral equation fixes the number of roots at $N/2$ and boundary complex roots can change it by one. Is there a description for a ground state with $N/2-k$ roots with finite $k$? Should it include some holes? Will it influence the corresponding correlation functions or it leads only to finite size corrections? I think that authors should add some comments to clarify these questions.

  3. There are some typos in the text for example on page 8 up an overall factor'' should be replaced byup to an overall factor''; in the first line of the Conclusion which non-longitudinal'' should be replaced bywith non-longitudinal'' etc., an additional proofreading can be useful.

---

## Round 1 · Referee Report · Anonymous (Referee 2) · 2023-3-14

Strengths

  1. Relevance - deals with a problem and set-up which has attracted attention in the literature.

  2. The main result in principle has numerous applications.

  3. Extensive literature review.

  4. Careful and precise definitions of all quantities used.

Weaknesses

  1. At times the paper is excessively technical.

  2. It is not clear how feasible it is to implement the main result in practice. For example for spin chains of some large but finite size, in a numerical set-up.

Report

The authors consider the problem of computing correlation functions in integrable spin chains with open boundary conditions in the separation of variables (SoV) approach.

The paper is another in a series by the authors. In previous works the authors computed correlation functions of local operators in the set-up where one of the boundary fields was longitudinal and the other was non-longitudinal. In the current article the authors consider a more general set-up where both boundaries are non-longitudinal but, rather than being completely generic, are related by a certain constraint. This constraint allows part of the spectrum of the model to be described by a homogeneous Baxter TQ equation and Bethe Ansatz equations. The authors refer to a conjecture in the literature that these equations also apply to the ground state at half filling. They then develop tools for computing correlation functions of bulk local operators which, under the assumption of the aforementioned conjecture, allows them to compute these correlation functions in the thermodynamic limit at zero temperature, which constitutes the main result of the article.

The main technical advancement is the “bulk-boundary decomposition” in the present set-up, which was initially developed by the authors and their collaborators in earlier work. Aided by this decomposition, bulk local operators are embedded into the SoV framework, and their finite-volume matrix elements are computed. Then, the thermodynamic limit is taken.

The result constitutes a major advancements in the computation of correlation functions in integrable systems with open boundary conditions. Owing to the relevance of such models in physical applications, the result is of particular importance. I recommend the article for publication.

I should however follow up on the point I listed in the "Weakness" category above. While I understand the authors desire for generality and precision, at times this is to the detriment of the reader. Indeed, propositions are proved in full generality in one go, while it might be more advantageous for the reader to see some simple set-ups considered first in order to get a feel for the results. This is also a common feature in other works of the authors. For example, Theorem 4.1 is an important result, but practically impossible to get a feel for what it is really saying in its current form, without spending time to work out all definitions and notations used, of which there are many. For the average reader who is not already an expert in the exact methods and notations and simply wants to have an overview of the results, it is unreasonable to expect them to do so. For example, presenting the results of Theorem 4.1 in the case m=1, M=2 before presenting the most general case would already help quite a bit.

At this point I do not think it feasible to implement such changes, as doing so would likely require a substantial amount of rewriting. However, I would ask that the authors consider such an approach in their future publications.

Requested changes

  1. The paper would benefit from a careful proof reading, while paying attention to English sentence structure, as sometimes it reads in an unnatural way to the point of somewhat obscuring the scientific content, which may in particular cause trouble for non-native speakers of English. For example "In [4] was also introduced for the study of these open spin chains a full algebraic formalism" -> "In [4] a complete algebraic formalism for the study of these open spin chains was introduced". "Reflexion" -> Reflection".

---

## Round 1 · Referee Report · Anonymous (Referee 3) · 2023-4-3

Strengths

  • New result about correlation functions for the open XXZ chain (with constraint)

  • Explicit boundary bulk decomposition formula for the gauged boundary Bethe state.

Weaknesses

  • involve only the usual TQ equation without the additional inhomogeneous term.

Report

The authors produce a new step torward the general integrable boundary correlation function. To overcome the lack of technique to study thermodynamical limit of the TQ equation with an additional inhomogeneous term, they impose one constraint. They perform the calculation of correlation functions for specific "local" operators and study the thermodynamic limit. The paper is well writed and deserve to be publish.

---

## Round 2 · Referee Report · Anonymous (Referee 1) · 2024-2-9

Report

Authors responded to my remarks, I recommend to accept the paper

---

## Round 2 · Referee Report · Anonymous (Referee 2) · 2024-2-10

Report

I am happy with the authors response to my report, and look forward to seeing concrete examples in their future publications. I recommend the paper for publication.

---

## Round 2 · Author Response

Dear Editor,

We would like to sincerely thank the three referees for their attentive study of our manuscript, for their requirements of clarifications, suggestions and list of grammar misprints.

In the new version of our manuscript, we have corrected the list of typos noticed by the referees, and implemented their grammar suggestions. We have also made some other minor corrections, see our detailed answer below.

1) On the referee 1 report.

  • About the request 1: We have modified the first paragraph of Section 4 to take into account the first referee request about the description of our basis of local operators. We have modified, in particular, the equation (4.1), where now we have defined the 4 local gauge transformed operators at a given site n (instead of their product). For clarity, we have also added, in footnote 7, their explicit expressions as linear combinations of the usual elementary operators at site n. Then, the elements of our basis (4.2) are simple tensor products of these local operators (4.1). The only peculiarity/complexity of our basis lays in the fact that the parameters b and bar b are functions by (4.3) of the full m-tuples of epsilons and prime epsilons. Following the referee request, we have also added a footnote where this dependence is written explicitly for m=1,2. One has to mention that our basis led to simple actions on the separate states which is indeed very similar to the ungauged action in our paper [20] (diagonal case), we have also added a footnote with this comment (footnote 6). This is a strong simplification, which enabled us to compute the action explicitly despite the combinatorial complexity induced by the use of the vertex-IRF transformation. This is of course the reason of the choice of this particular basis.

  • About the request 2: It is clear that we are not doing a full analysis of the ground state of the boundary spin chain here. We have made this even more transparent by modifying one sentence in the paragraph after formula (6.7) and adding an additional sentence in the Conclusion. Such an analysis is quite involved, since we have to distinguish many cases according to the different values of the boundary parameters, and we have even to compare what is happening in two different sectors if we trust the completeness conjecture of [14]: see for instance the analysis done in [41] for the diagonal case, knowing that here it is much more involved since there are much more cases to distinguish. Such an analysis of the ground state, if done properly and precisely, would probably deserve a separate publication. Therefore, we are doing some assumptions here: as stated in page 19, we assume that we are in a configuration of Bethe roots such that the ground state is indeed described by an infinite number of real Bethe roots, leading to a density function (6.4) solution of (6.5) and (6.6), with possibly some additional boundary roots of the form (6.7). Under this assumption we obtain our result (6.8). We nevertheless point out just before these assumptions that the thermodynamic analysis of the ground state can in principle be implemented as in [8,9,41], if the latter is in the sector N/2, or even in a sector which differs from N/2 by a finite number: this will lead to the ground state density function (6.4). Of course, depending on the precise sector and on the value of the boundary parameters, the fine structure of the ground state may be very different (with possibly, among other possible effects, the presence of some holes, as mentioned by Referee 2). The presence of a finite number of holes should however not modify the leading order of the result in the thermodynamic limit, and we have added a footnote (footnote 12) mentioning it. In principle, the presence of a finite number of additional roots should not modify the leading order either, except if these roots correspond (in the thermodynamic limit) to some singularities of our expression: this is precisely what happens for the boundary roots of the type (6.7), and this is why they are especially important for the statement of our result, as in the diagonal case [20,41]. The first paragraph of section 6, instead, is meant to clarify that, under the completeness conjecture of [14], we have that if M is of order N/2 then the same is true for M’=N-1-M so that the ground state has a number of roots of order N/2, independently if it is in the sector M or M’, and so the ground state analysis which is sketched just after that applies.

We have slightly modified a few sentences (and added footnote 12) in these paragraphs so as to make all this clearer.

2) On the referee 2 report.

We understand the referee statement about technical complexities of our manuscript, although this is somehow inevitable due to the technical complexity of the research subject. We will try to make our best, in our future publications, to simplify somewhere the exposition by giving concrete examples.

About the referee comment on Theorem 4.1, we would like to mention that the form of this result mimics similar results that have been previously obtained for the correlations functions of the spin chain with different types of boundary conditions. As mentioned in Remark 2, it has of course the same formal form as in our previous work [1] with the special boundary condition (4.17). More remarkably, it has mainly the same structure as in Proposition 5.2 of [20], which stated the action of the natural basis of local operators on usual (ungauged) Bethe vectors for diagonal boundary conditions. Here, the main difference is in the last two lines of (4.12), when compared to (5.15) of [20], where we can read of explicitly the dependence from the b- and bar b-parameters of our current local basis. It is this relative simplicity that allows us to compute correlation functions in our current general boundary conditions.

We also agree that it is not clear how much practical may be our formulae (and our final result) for implementing a numerical set-up. This is only the first step towards a full understanding of the correlation functions of the model. Nevertheless, considering the complexity of the problem of dealing with this kind of boundary conditions, we think it is important to understand first how to generalize the results which are known in the diagonal case [20] before trying to go further, and this is the purpose of the present work. A possible further and more concrete study would be for instance to try to generalize the results of [41] to this case. For more general numerical applications, the difficulty in the open chain (even for diagonal boundary conditions) is that, contrary to the periodic case [93], we lack a compact and manageable representation for the form factors as soon as we go away from the boundary. Trying to overcome this difficulty is an interesting and very challenging problem, but it should obviously be addressed first in the simpler diagonal case.

3) On the referee 3 report.

The referee 3 points out as a weakness the fact that we restrain ourselves to the standard TQ-equation. In fact, as the referee mentions, our main motivation to consider the case with one constrain is due to the fact that the problem can be reduced to the standard TQ-equation for which the thermodynamic analysis can be addressed on more standard basis. Instead, the unconstraint most general integrable boundary conditions naturally lead to a TQ-equation with one additional (inhomogeneous) term, once one asks for trigonometric polynomial Q-functions. There exist some analyses of this most general case, but it is clearly much more difficult to deal with than the constraint case. In particular, it is not clear how to deal with the thermodynamic limit in that case, since the Bethe roots can a priori not be described by a simple density function as in the constraint case. But we agree with the referee that considering this most general case is an interesting – although difficult – open problem, which is commented as such in the conclusion.

---

## Round 2 · List of Changes

• small grammatical corrections, as requested by the referees
  • slight modifications at the beginning of section 4 so as to better clarify the definition of the basis of local operators (4.2), as requested by referee 1
  • slight modifications of section 6, just after (6.3) and around (6.7), and one sentence added in the conclusion, see our detailed answer to the request 2 of referee 1

---

## Editorial Decision

published